# Functional implications of the exon 9 splice insert in GluK1 kainate receptors

**Surbhi Dhingra[1], Prachi M Chopade[1†], Rajesh Vinnakota[1†], Janesh Kumar[1,2]\***

[1]Laboratory of Membrane Protein Biology, National Centre for Cell Science, Pune, India; [2]Laboratory of Membrane Protein Biology, CSIR-Centre for Cellular and Molecular Biology, Hyderabad, India

## eLife Assessment

This **important** study shows that a splice variant of the kainate receptor Glu1-1a that inserts 15 amino acids in the extracellular N-terminal region substantially changes the channel's desensitization properties, the sensitivity to glutamate and kainate, and the effects of modulatory Neto proteins. In the revised paper the authors have clarified several points raised by reviewers but the structural portion of the study has not been improved and consequently, more data are needed to determine the molecular mechanism by which the insert changes the functional profile of the channel. Even so, these **solid** findings advance our understanding of splice variants among glutamate receptors and will be of interest to neuro- and cell-biologists and biophysicists in the field.

**\*For correspondence:**
janesh@ccmb.res.in

[†]These authors contributed equally to this work

**Competing interest:** The authors declare that no competing interests exist.

**Abstract** Kainate receptors are key modulators of synaptic transmission and plasticity in the central nervous system. Different kainate receptor isoforms with distinct spatiotemporal expressions have been identified in the brain. The GluK1-1 splice variant receptors, which are abundant in the adult brain, have an extra fifteen amino acids inserted in the amino-terminal domain (ATD) of the receptor resulting from alternative splicing of exon 9. However, the functional implications of this post-transcriptional modification are not yet clear. We employed a multi-pronged approach using cryogenic electron microscopy, electrophysiology, and other biophysical and biochemical tools to understand the structural and functional impact of this splice insert in the extracellular domain of GluK1 receptors. Our study reveals that the splice insert alters the key gating properties of GluK1 receptors and their modulation by the cognate auxiliary Neuropilin and tolloid-like (Neto) proteins 1 and 2. Mutational analysis identified the role of crucial splice residues that influence receptor properties and their modulation. Furthermore, the cryoEM structure of the variant shows that the presence of exon 9 in GluK1 does not affect the receptor architecture or domain arrangement in the desensitized state. Our study thus provides the first detailed structural and functional characterization of GluK1-1a receptors, highlighting the role of the splice insert in modulating receptor properties and their modulation.

## Introduction

Kainate receptors (KARs), a subfamily of ionotropic glutamate receptors (iGluRs), are required in the vertebrate brain for postsynaptic neurotransmission and presynaptic regulation of transmitter release (*Erreger et al., 2004*; *Huettner, 2003*; *Lerma, 2003*; *Pinheiro and Mulle, 2006*). They are known to mediate characteristic small-amplitude excitatory postsynaptic currents (EPSC) with slow kinetics in the hippocampal regions of the central nervous system compared to their counterparts, α-amino-3-hydroxy-5-methyl-4-isoxazole propionic acid (AMPA) and N-methyl-D-aspartate (NMDA) receptors (*Castillo et al., 1997*; *Lerma et al., 2001*; *Traynelis et al., 2010*). Furthermore, they play a vital role in

the maturation of neuronal circuits during development by interacting with G proteins (*Lerma, 2003*; *Lerma and Marques, 2013*). Any functional defect predisposes the brain to various disorders, such as autism, epilepsy, schizophrenia, and neuropathic pain (*Lerma and Marques, 2013*; *Bowie, 2008*; *Valbuena and Lerma, 2021*).

Kainate receptors are composed of subunits from two gene families: low kainate affinity and high kainate affinity. The first gene family includes GluK1-GluK3, which can form both homomeric and heteromeric receptors with subunits from the same family or from the high-kainate affinity family. On the other hand, GluK4-GluK5 are high-affinity subunits that must assemble with low-affinity subunits to create functional receptors. The assembly of different subunits results in the formation of various receptor configurations, which contribute to the wide range of functional properties observed in kainate receptors.

The functional repertoire of kainate receptors is further enhanced by RNA editing (*Barbon and Barlati, 2011*; *Seeburg, 2002*) and alternative splicing (*Jaskolski et al., 2004*). In particular, GluK1-containing KARs mainly present in the hippocampus, cortical interneurons, Purkinje cells, and sensory neurons undergo alternative splicing and RNA editing (*Bernard and Khrestchatisky, 1994*). Alternative splicing of the C-terminal domain that produces four isoforms, GluK1-a (shortest C-terminal), GluK1-b, GluK1-c, and GluK1-d (*Bettler et al., 1990*; *Gregor et al., 1993*; *Pinheiro and Mulle, 2006*; *Sommer et al., 1992*) has been studied (*Ren et al., 2003*). However, the functional impact of the mRNA splicing of the N-terminal domain (exon 9) resulting in isoforms GluK1-1 and GluK1-2, with GluK1-1 containing an additional 15 amino acids in the amino-terminal domain (ATD) is not understood. Interestingly, the splice junctions for GluK1-1 were reported to be more frequent than those for GluK1-2, and both variants showed similar expression in different species (*Herbrechter et al., 2021*). This 15-amino acid insertion in the ATD is exclusive to the GluK1 subunit and may impart unique properties to KARs containing this variant. Therefore, investigations of functional differences between GluK1 receptors with and without the 15 amino acid splice insert are necessary but missing in the field. Furthermore, the impact of the N-terminal splice insert in GluK1-1 receptors on its modulation by cognate auxiliary subunits (*Fisher, 2015*; *He et al., 2021*; *Sheng et al., 2015*; *Vinnakota et al., 2021*), neuropilin and tolloid-like (Neto) proteins is also unknown.

Therefore, we conducted structure-function studies to understand the mechanisms of GluK1-1a receptor function and the effects of the ATD splice on receptor assembly, stability, and modulation by Neto proteins. Whole-cell and excised outside-out patch-clamp-based functional assays were performed to investigate the differences between the ATD splice variants GluK1-1a (exon 9) and GluK1-2a (lacking exon 9) and their modulation by Neto proteins. Furthermore, mutational analysis was performed to identify important splice residues that affect receptor functions and their modulation by Neto proteins. Additionally, we expressed and purified rat GluK1-1a from HEK293 GnTI- cells and determined the single-particle cryo-EM structures of the receptors trapped in the desensitized state to understand the effect of splice on receptor architecture.

## Results

### Spatiotemporal expression pattern of GluK1-1 in the brain

Kainate receptor subunits are differentially and spatially regulated in vertebrates. We analyzed publicly available human transcriptomics data to determine whether the exon encoding the GluK1 ATD splice (exon 9) was differentially expressed in the human brain. RNASeq data analysis of *GRIK1* (Ensembl ID: ENSG00000171189) collected from the BrainSpan ATLAS indicated that exon 9 is present in multiple brain areas that also show prominent GluK1 expression. GRIK1 exon 9 expression varies significantly across brain regions and developmental stages, with dynamic patterns indicating its crucial role in brain development. High expression is noted during critical periods such as early embryonic, postnatal, and childhood stages, particularly in regions like the cortical plate (CP), hippocampus (HIP), amygdala (AMY) and striatum (STR) (*Figure 1*; *Figure 1—figure supplement 1*). These patterns suggest GRIK1 exon 9's importance in the functional maturation of these brain regions. For example, in the cerebellar cortex, GRIK1 exon 9 expression peaks during early development and childhood and stabilizes into adulthood, underscoring its role in cortical development and function. (*Figure 1—figure supplement 1*). The significant variability in exon 9 expression across different brain regions, along with its potential role in brain development and functional maturation, motivated us to investigate the underlying

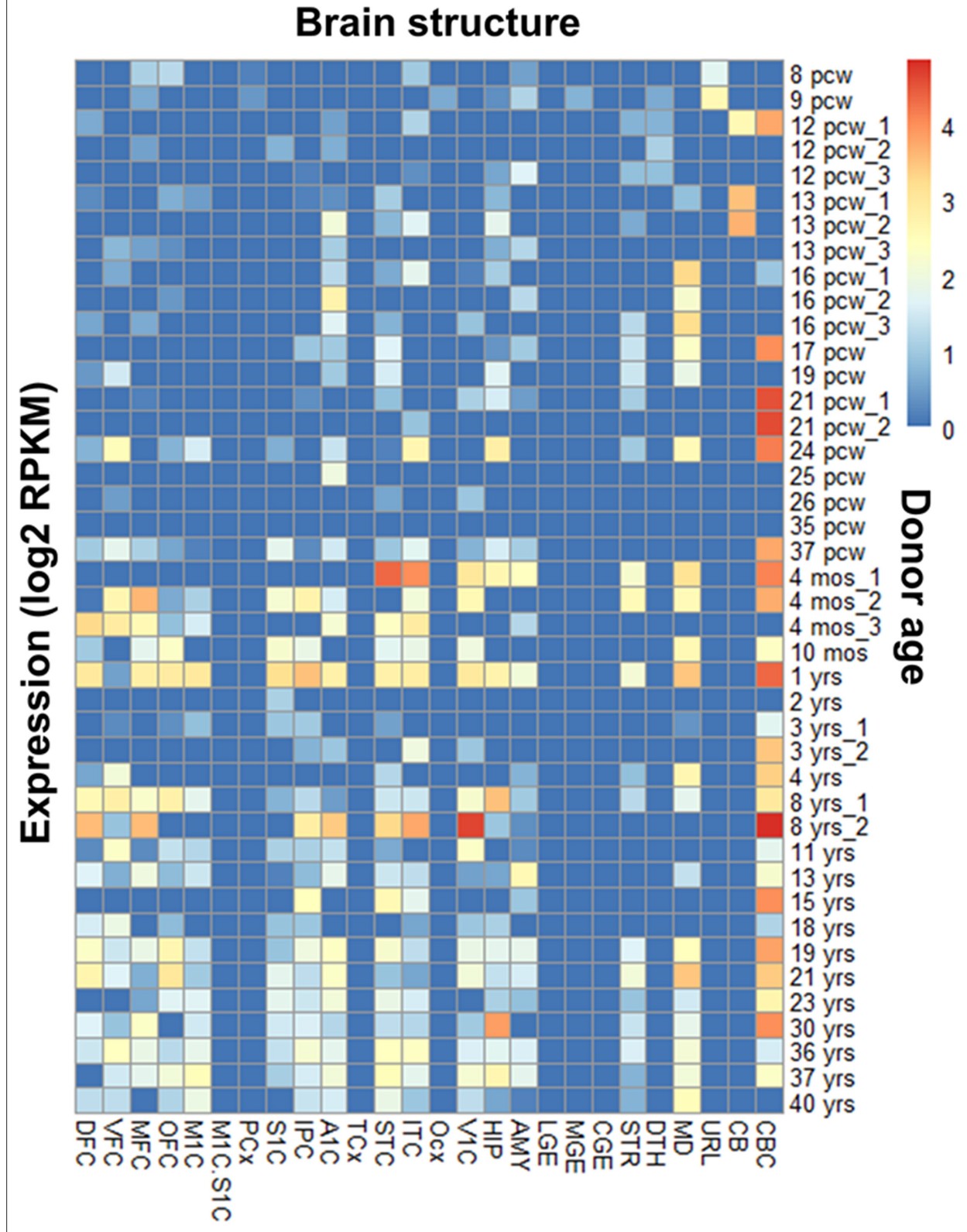

**Figure 1.** RNA-seq analysis (BrainSpan atlas) demonstrates an abundance of *GRIK1* exon 9 in the human brain. The heat map shows the presence of exon 9 (45 bp; ENSE00001313812) that codes for GluK1 amino-terminal domain (ATD) splice in different brain regions from the embryonic to adult stage. Exon 9 expression coincides with well-studied areas for the GRIK1 gene like the cerebellar cortex, visual cortex, etc. Various regions of the brain and the donor age are represented on the x-axis and y-axis, respectively. The donor age has been abbreviated as pcw (post-conception weeks), mos

*Figure 1 continued on next page*

*Figure 1 continued*

(months), and yrs (years). The regions of the human brain are abbreviated as: DFC (dorsolateral prefrontal cortex), VFC (ventrolateral prefrontal cortex), MFC (anterior [rostral] cingulate [medial prefrontal] cortex), OFC (orbital frontal cortex), M1C (primary motor cortex area M1, area 4), M1C.S1C (primary motor-sensory cortex [samples]), PCx (parietal neocortex), S1C primary somatosensory cortex (area S1, areas 3,1,2), IPC (posteroventral [inferior] parietal cortex), A1C (primary auditory cortex core), A1C (primary auditory cortex [core]), TCx (temporal neocortex), STC (posterior [caudal] superior temporal cortex, area 22 c), ITC (inferolateral temporal cortex) (area TEv, area 20), OCx (occipital neocortex), V1C primary visual cortex (striate cortex, area V1/17), HIP (hippocampus), AMY (amygdaloid complex), LGE (lateral ganglionic eminence), MGE (medial ganglionic eminence), CGE (caudal ganglionic eminence), STR (striatum), DTH (dorsal thalamus), MD (mediodorsal nucleus of thalamus), URL (upper [rostral] rhombic lip), CB (cerebellum), and CBC (cerebellar cortex). Blue and red color indicates zero and maximum expression, respectively.

The online version of this article includes the following source data and figure supplement(s) for figure 1:

**Source data 1.** RNA-seq analysis data for GRIK1 exon 9.

**Figure supplement 1.** Representative RNA-seq analysis (BrainSpan atlas) of the cerebellar cortex (CBC) demonstrates the high expression of exon 9 in the human brain.

**Figure supplement 1—source data 1.** RNA-seq analysis data for GRIK1 exon 9 for cerebellar cortex (CBC).

molecular mechanisms. This prompted us to perform a detailed structure-function analysis of the GluK1-1a splice variant.

## The ATD splice imparts functional diversity to GluK1 receptors

Previous studies on functional analysis of GluK1 receptors have primarily focused on GluK1-2 isoform that lacks the ATD splice insert. Therefore, we conducted an extensive electrophysiological investigation to assay the functional differences between GluK1-1a and GluK1-2a. The only difference between these two variants is the presence of 15 amino acids (KASGEVSKHLYKVWK) insertion in the R2 subdomain (lower lobe) of the GluK1-1a ATD. We evaluated multiple gating properties of the two variants by whole-cell patch-clamp electrophysiology. Interestingly, we observed that GluK1-1a receptors desensitize significantly slowly when compared to GluK1-2a ($\tau_{Des}$, GluK1-1a: 5.21±0.50 ms; GluK1-2a: 3.55±0.23 ms **p=0.0092) on prolonged treatment with 10 mM glutamate (*Figure 2A*; *Table 1*). We also tested receptor desensitization by applying saturating concentrations of kainate (1 mM). Since the receptors displayed prolonged desensitizing kainate currents, instead of the rate ($\tau_{Des}$), we calculated the % desensitization values at 1 s for comparison. We observed that the % desensitization with kainate was distinct for both ATD splice variants with significantly slower desensitization in the variant with the splice insert (GluK1-1a: 72.06±2.33%, GluK1-2a: 93.2±0.55 ***p=0.0006) consistent with the glutamate evoked currents (*Figure 2B*; *Table 1*).

Next, we evaluated glutamate sensitivity for the two variants. Dose-response experiments (glutamate, GluK1-1a: 0.1–2 mM, and GluK1-2a: 0.01–3 mM) showed a significantly lower potency of glutamate for GluK1-1a variant compared to the non-spliced form ($EC_{50}$ glutamate), GluK1-1a: 379.3±52 µM, GluK1-2a: 187.7±33 µM *p=0.0129 (*Figure 2C*; *Table 1*). However, for high-affinity agonist kainate, the dose-response curves for both variants were similar, likely indicating differences in the stability of glutamate *versus* the kainate-bound states in GluK1-1a and GluK1-2a (*Figure 2D*; *Figure 2—figure supplement 1*). Thus, the potency of kainate (1 mM) *versus* glutamate (10 mM) ($I_K/I_G$ ratio) is significantly higher for GluK1-1a compared to GluK1-2a (GluK1-1a: 1.51±0.13, GluK1-2a: 0.56±0.4 ****p<0.0001) (*Figure 2D*; *Table 1*).

Furthermore, we investigated the voltage-dependent endogenous polyamine block and found significant differences between the two variants. The presence of splice residues seemed to enhance outward currents at positive potentials in GluK1-1a compared to GluK1-2a (rectification index,+90 mV/–90 mV; GluK1-1a=0.96 ± 0.11; GluK1-2a=0.61 ± 0.10 *p=0.0385) without affecting the reversal potential (*Figure 2E*; *Table 1*). How splice residues situated ~92 Å away from the TM domain (distance between atoms W381 CA in the ATD and L636 CA in the TM3) affect the pore properties is unclear. Earlier reports suggest that rectification in KARs is mainly affected by the TM2 region (*Bowie and Mayer, 1995*). However, a recent report in which the ATD of GluK2 was deleted also showed enhanced rectification (*Li et al., 2019*). Splice residues likely alter pore structure by allosteric mechanisms that have (are) yet to be identified, thereby affecting rectification (*Perrais et al., 2009*). We also performed outside-out patch recordings to examine GluK1-1a receptors. However, we observed extremely weak electrical currents with low amplitudes when GluK1-1a was expressed in

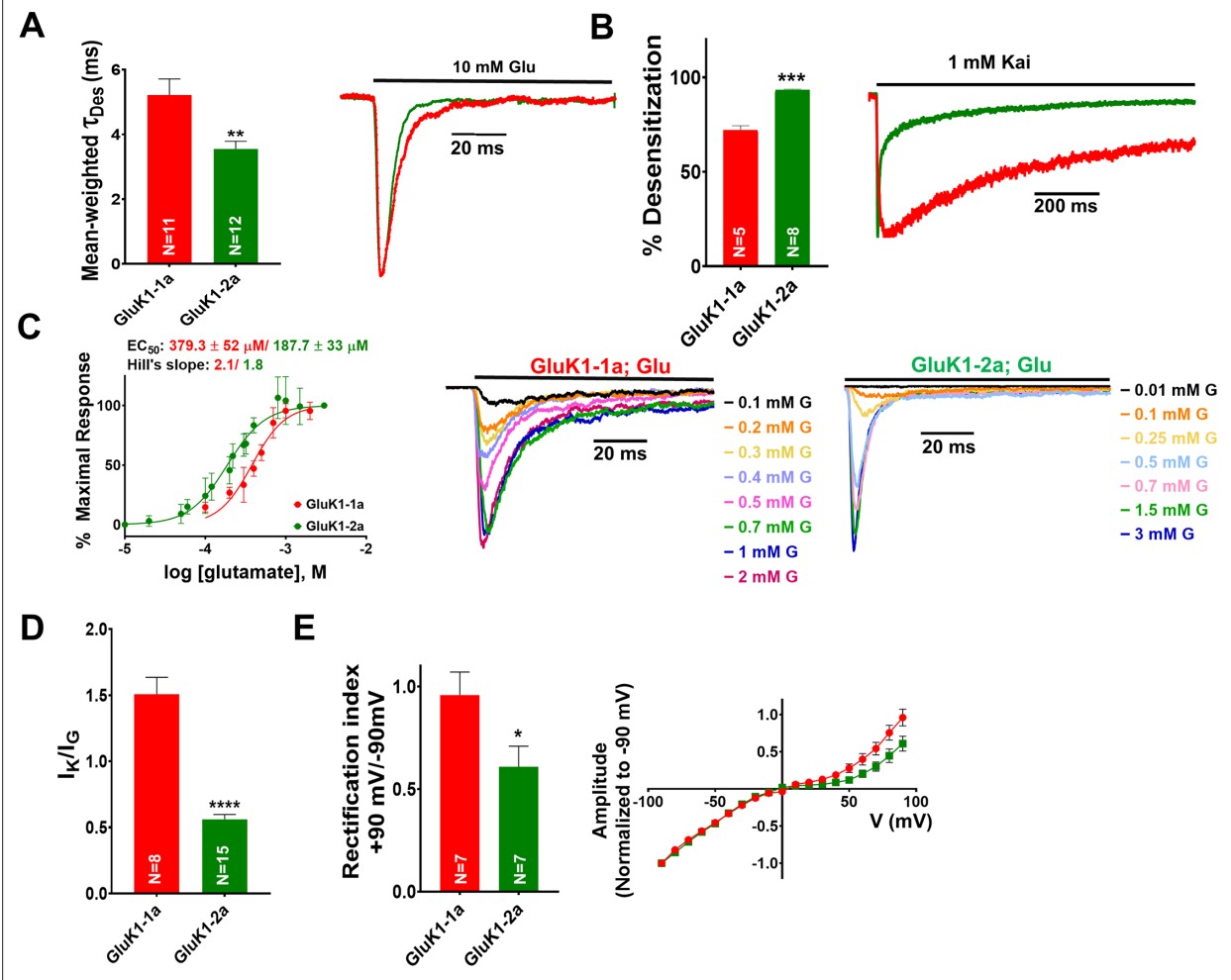

**Figure 2.** Amino-terminal domain (ATD) splice insert affects the gating properties of the GluK1-1a homomeric receptors. (**A**) Displays mean-weighted Tau ($\tau_{Des}$) values for GluK1-1a wild-type (red) and GluK1-2a wild-type (green) in the presence of glutamate. The inset shows representative normalized traces for GluK1-1a and GluK1-2a with 10 mM glutamate. (**B**) Displays the percent desensitization values calculated at 1 s for GluK1-1a wild-type (red) and GluK1-2a wild-type (green) in the presence of kainate. Representative normalized traces for GluK1-1a and GluK1-2a with 1 mM kainate are shown. (**C**) Demonstrates glutamate dose-response curves for GluK1-1a and GluK1-2a. Representative aligned traces for both receptors at various glutamate concentrations are shown. The kainate dose responses for the splice variants are shown in *Figure 2—figure supplement 1*. (**D**) The ratio of currents evoked by kainate and glutamate is plotted for GluK1-1a and GluK1-2a. (**E**) The ratio of currents evoked by the application of 10 mM glutamate at +90 mV and –90 mV for the GluK1-1a and GluK1-2a receptors is shown. Representative IV plots are depicted for GluK1-1a and GluK1-2a for the entire voltage ramp (–90 to +90 mV). Error bars indicate mean ± SEM, N in each bar represents the number of cells used for analysis, and * indicates the significance at a 95% confidence interval.

The online version of this article includes the following source data and figure supplement(s) for figure 2:

**Source data 1.** Data used for the electrophysiology plots.

**Figure supplement 1.** Kainate-evoked responses for GluK1 receptors.

**Figure supplement 1—source data 1.** Data used for the electrophysiology plots.

---

isolation. Consequently, the data we obtained from these recordings did not provide reliable results for curve fitting or thorough analysis.

## ATD splice insert impacts GluK1 receptor modulation by Neto proteins

Neto 1 and Neto 2 proteins significantly influence the surface expression, synaptic localization, and functional properties of the GluK1-2a receptors (*Copits et al., 2011*; *Palacios-Filardo et al., 2016*; *Sheng et al., 2015*). It has been demonstrated that desensitization of GluK1-2a is accelerated by Neto1 but delayed by Neto2 (*Sheng et al., 2015*; *Copits et al., 2011*; *Palacios-Filardo et al., 2016*).

---

**Table 1.** Whole-cell patch clamp recordings of GluK1-1a, GluK1-1a$_{EM}$, GluK1-2a, and GluK1-1a mutants in the absence or presence of Neto1 (green) or Neto2 (peach).

Errors are reported as SEM. Statistical significance is reported at 95 % CI. $p<0.05$ (*), $p<0.01$ (**), $p<0.001$ (***), $p<0.0001$ (****) for comparisons between wild-type GluK1-1a receptor with EM construct, GluK1-2a, or various mutants in presence or absence of Neto proteins.

'a' denotes the lack of rectification index value due to no conductance observed at positive potentials in the K$_{368}$-E mutant alone.

| S. No. | Name of the construct | Desensitization | | I$_K$/I$_G$ | Recovery ($\tau_{Rec}$ in s) | Rectification (+90 mV/–90 mV) | Total No. of cells tested |
|---|---|---|---|---|---|---|---|
| | | 10 mM G ($\tau_{Des}$ in ms) | 1 mM K (%) | | | | |
| 1 | GluK1-1a$_{EM}$ | 1.94+0.13 N=8 (****p<0.0001) | - | - | 2.63±0.04 N=3 (p=.3501) | - | 23 |
| 2 | GluK1-1a | 5.21±0.50 N=11 | 72.06±2.33 N=5 | 1.51±0.13 N=8 | 3.53±0.81 N=4 | 0.96±0.11 N=7 | 19 |
| | +Neto1 | 3.56±0.22 N=9 (**p=0.0090) | 49.61±5.23 N=7 (**p=0.0075) | 1.25±0.04 N=8 (p=0.1500) | 0.68±0.07 N=8 (*p=0.0390) | 1.16±0.09 N=7 (p=0.1880) | 11 |
| | +Neto2 | 69.62±9.98 N=6 (**p=0.0024) | 68.96±4.36 N=9 (p=0.7459) | 1.00±0.07 N=13 (**p=0.0096) | 8.32±0.81 N=5 (**p=0.0044) | 0.80±0.05 N=4 (p=0.2401) | 17 |
| 3 | GluK1-2a | 3.55±0.23 N=12 (**p=0.0092) | 93.20±0.55 N=8 (***p=0.0006) | 0.56±0.04 N=15 (****p<0.0001) | 5.31±0.50 N=7 (p=0.1191) | 0.61±0.10 N=7 (*p=0.0385) | 21 |
| | +Neto1 | 4.32±0.34 N=8 (p=0.1495) | 76.99±3.41 N=6 (**p=0.0085) | 1.37±0.12 N=10 (****p<0.0001) | 1.15±0.12 N=5 (***p=0.0002) | 1.14±0.14 N=3 (*p=0.0338) | 14 |
| | +Neto2 | 21.68±2.64 N=6 (**p=0.0017) | 65.98±2.41 N=7 (****p<0.0001) | 1.16±0.04 N=8 (****p<0.0001) | 7.91±0.71 N=4 (*p=0.0430) | 1.37±0.10 N=3 (**p=0.0022) | 18 |
| 4 | K$_{375}$H$_{376}$-A | 3.80±0.33 N=5 (*p=0.0332) | 71.07±5.27 N=5 (p=0.9998) | 1.43±0.07 N=5 (p=0.6195) | 1.10±0.22 N=4 (**p=0.0095) | 0.78±0.05 N=4 (p>0.9999) | 9 |
| | +Neto1 | 3.38±0.14 N=8 (p=0.4947) | 54.64±5.18 N=6 (p=0.5086) | 1.76±0.14 N=8 (**p=0.0070) | 1.15±0.11 N=9 (**p=0.0026) | 0.64±0.04 N=4 (***p=0.0008) | 10 |
| | +Neto2 | 42.57±11.96 N=4 (p=0.1280) | 61.03±5.12 N=5 (p=0.6416) | 1.02±0.02 N=5 (Pp0.7791) | 9.32±0.15 N=3 (p=0.6286) | 0.60±0.06 N=5 (p=0.1138) | 6 |
| 5 | K$_{375/379/382}$-A | Low peak amplitudes (<40 pA) | | | | | 15 |
| | +Neto1 | 2.82±0.18 N=6 | 64.13±5.26 N=5 | 1.42±0.08 N=6 | 0.74±0.08 N=6 | 1.28±0.10 N=6 | 8 |

*Table 1 continued on next page*

*Table 1 continued*

| S. No. | Name of the construct | Desensitization | 10 mM G ($\tau_{Des}$ in ms) | 1 mM K (%) | $I_K/I_G$ | Recovery ($\tau_{Rec}$ in s) | Rectification (+90 mV/−90 mV) | Total No. of cells tested |
|---|---|---|---|---|---|---|---|---|
| | +Neto2 | Low peak amplitudes (≤40 pA) | | | | | | 27 |
| 6 | $Y_{378}V_{380}W_{381}$-A | Low peak amplitudes (≤40 pA) | | | | | | 18 |
| | +Neto1 | | 3.31±0.16 N=9 | 71.48±4.66 N=5 | 1.76±0.21 N=7 | 1.02±0.07 N=7 | 0.77±0.14 N=5 | 10 |
| | +Neto2 | | 45.79±17.32 N=3 | 51.82±4.88 N=3 | 1.19±0.11 N=3 | 9.45±1.21 N=3 | 1.03±0.44 N=4 | 22 |
| 7 | $K_{375}$-$K_{382}$-8A | Non-functional N=15 | | Non-functional N=5 | - | - | Non-functional N=5 | 15 |
| | +Neto1 | Non-functional N=11 | | Non-functional N=4 | - | - | Non-functional N=5 | 11 |
| | +Neto2 | Non-functional N=11 | | Non-functional N=5 | - | - | Non-functional N=6 | 11 |
| 8 | $K_{375}H_{376}$-E | Low peak amplitudes (≤40 pA) | | | | | | 13 |
| | +Neto1 | | 3.59±0.29 N=4 | 13.78±3.16 N=5 | 0.70±0.19 N=4 | 0.63±0.10 N=3 | 0.68±0.19 N=3 | 23 |
| | +Neto2 | Low peak amplitudes (≤40 pA) | | | | | | 36 |
| 9 | $H_{376}$-E | | 5.07±0.42 N=7 (p=0.8408) | 29.80±13.21 N=4 (*p=0.0471) | 0.88±0.05 N=3 (*P=0.0272) | 1.48±0.18 N=8 (*p=0.0112) | 0.44±0.07 N=6 (**p=0.0023) | 19 |
| | +Neto1 | | 2.69±0.15 N=5 (**p=0.0065) | 59.24±8.62 N=4 (p=0.3813) | 2.36±0.30 N=4 (*P=0.0328) | 0.66±0.05 N=5 (p=0.7939) | 0.80±0.19 N=4 (p=0.3625) | 6 |
| | +Neto2 | | 50.88±6.49 N=4 | 28.82±6.43 N=5 (**p=0.0034) | 1.43±0.13 N=4 (*P=0.0347) | 4.98±0.81 N=3 (*P=0.0237) | 0.42±0.08 N=8 (**p=0.0056) | 50 |
| 10 | $K_{375/379/382}H_{376}$-E | | 9.62±1.47 N=6 (*p=0.0290) | 54.25±8.28 N=3 (*p=0.0396) | 1.17±0.28 N=3 (P=0.3733) | 4.83±0.31 N=3 (p=0.2774) | 0.62±0.14 N=7 (*p=0.0499) | 36 |
| | +Neto1 | | 3.74±0.20 N=8 (p=0.5390) | 57.72±12.65 N=3 (p=0.5996) | 1.69±0.15 N=6 (*p=0.0326) | 0.66±0.08 N=8 (p=0.8702) | 1.23±0.07 N=6 (p=0.5351) | 9 |
| | +Neto2 | | 65.57±17.04 N=8 (p=0.8412) | 26.66±2.05 N=4 (****p<0.0001) | 0.86±0.08 N=4 (p=0.2045) | 6.57±0.61 N=3 (p=0.3588) | 0.72±0.09 N=4 (p=0.6677) | 48 |
| 11 | $K_{368/375/379/382}H_{376}$-E | Low peak amplitudes (≤40 pA) | | | | | | 20 |

*Table 1 continued*

| S. No. | Name of the construct | Desensitization | | | | | | Total No. of cells tested |
|---|---|---|---|---|---|---|---|---|
| | | 10 mM G ($\tau_{Des}$ in ms) | 1 mM K (%) | $I_K/I_G$ | Recovery ($\tau_{Rec}$ in s) | Rectification (+90 mV/–90 mV) | | |
| | +Neto1 | 4.86±0.67 N=5 (p=0.1242) | 10.39±.3.53 N=5 | 1.15±0.20 N=4 (p=0.6495) | 0.21±0.12 N=3 | 0.59±0.13 N=5 (**p=0.0081) | | 21 |
| | +Neto2 | Low peak amplitudes (≤40 pA) | | | | | | 22 |
| 12 | K$_{368}$-E | 7.89±1.14 N=3 (*p=0.0334) | 46.66±5.88 N=5 (**p=0.0093) | 0.81±0.13 N=5 (**p=0.0037) | 5.61±0.95 N=3 (*p=0.0417) | a N=7 (****p<0.0001) | | 32 |
| | +Neto1 | 4.37±0.25 N=4 (*p=0.0393) | 59.68±8.72 N=3 (p=0.3851) | 6.99±0.47 N=5 (***p=0.0002) | 0.43±0.05 N=6 (*p=0.0118) | 2.84±0.16 N=3 (**p=0.0021) | | 8 |
| | +Neto2 | 79.40±4.44 N=5 (p=0.4012) | 33.47±8.98 N=5 (*p=0.0388) | 2.13±0.13 N=4 (***p=0.0007) | 7.66±0.83 N=3 (p=0.9473) | 0.76±0.26 N=8 (p=0.9860) | | 43 |

Hence, to understand the influence of these KAR auxiliary proteins on GluK1-1a, we performed an electrophysiological analysis of GluK1-1a and GluK1-2a receptors co-expressed with either Neto1 or Neto2. Co-expression of Neto1 hastened desensitization of GluK1-1a significantly but not of GluK1-2a at saturating glutamate concentrations ($\tau_{Des}$, GluK1-1a +Neto1: 3.56±0.22 ms **p=0.0090; GluK1-2a + Neto1: 4.32±0.34 p=0.1495). On the other hand, Neto2 led to a ~13.4 fold decrease in the desensitization rate of GluK1-1a ($\tau_{Des}$, GluK1-1a: 5.21±0.50 ms, +Neto2: 69.62±9.98 ms **p=0.0024) while the desensitization rate of GluK1-2a was decreased only by ~6.1 fold ($\tau_{Des}$, GluK1-2a: 3.55±0.23 ms, +Neto2: 21.68±2.64 ms **p=0.0017). Thus, while Neto1 accelerated the desensitization of GluK1-1a, Neto2 significantly slowed it at saturating glutamate concentrations. The rate of desensitization for GluK1-1a was approximately 3.2 times slower compared to GluK1-2a (***p=0.0009) when coexpressed with Neto2 (*Figure 3A*; *Table 1*). Thus, the presence of splice insert leads to differential modulation of GluK1 desensitization by Neto proteins.

Moreover, Neto1 also enhanced the recovery from desensitization for both the variants ($\tau_{Recovery}$, GluK1-1a: 3.53±0.81 s, +Neto1: 0.68±0.07 s *p=0.0390; GluK1-2a: 5.31±0.50 s +Neto1: 1.15±0.12 s ***p=0.0002). GluK1-1a recovers ~1.7 times faster than GluK1-2a (*p=0.0125) when co-expressed with Neto1. Neto 2, on the other hand, slowed recovery for both variants to a similar extent ($\tau_{Recovery}$, GluK1-1a: 3.53±0.81 s, +Neto2: 8.32±0.81 s **p=0.0044; GluK1-2a: 5.31±0.50 s, +Neto2: 7.91±0.71 s *p=0.0430) and did not show differential modulation (*Figure 3B*; *Table 1*). In addition, both Neto1 and Neto2 increased the potency of glutamate for GluK1-1a by 9.7-fold (39±10 μM) and 11.2-fold (34±8 μM), respectively (*Figure 3C*; *Figure 3—figure supplement 1A*). This is similar to the effect observed in GluK1-2a receptors whereby the glutamate $EC_{50}$ was shown to increase by Neto proteins Neto1: 34-fold and Neto2: 7.5-fold (*Palacios-Filardo et al., 2016*) and Neto1/2: 30–10 X (*Fisher, 2015*).

Since we observed a significant increase in the $I_K/I_G$ ratios in GluK1 due to the presence of splice insert, we next aimed to determine the influence of co-expressing Neto proteins on this parameter. Interestingly, while Neto 1 and Neto2 reduced the $I_K/I_G$ ratios in GluK1-1a (GluK1-1a: 1.51±0.13, +Neto1: 1.25±0.04 p=0.1500, +Neto2: 1.0±0.07 **p=0.0096), they increased it for the non-spliced variant (GluK1-2a: 0.56±0.04, +Neto1: 1.37±0.12 ****p<0.0001, +Neto2: 1.16±0.04 ****p<0.0001) (*Figure 3D*; *Figure 3—figure supplement 1B*; *Table 1*, *Zhang et al., 2009*) highlighting the effect of splice residues. Interestingly, this differential modulation of the two variants by Neto proteins resulted in comparable $I_K/I_G$ ratios.

Next, we investigated the effects of Neto1 and Neto2 on the voltage-dependent endogenous polyamine block since the presence of splice insert had enhanced the outward rectification of GluK1. We observed that both Neto1 and Neto2 significantly enhanced the outward rectification of GluK1-2a (GluK1-2a: 0.61±0.10, +Neto1: 1.14±0.14 *p=0.0338, +Neto2: 1.37±0.10 **p=0.0022). However, they did not significantly increase it for GluK1-1a (GluK1-1a: 0.96±0.11, +Neto1: 1.16±0.09 p=0.1880, +Neto2: 0.80±0.05 p=0.2401) and did not show differential modulation (*Figure 3E and F*; *Table 1*).

We also calculated desensitization and deactivation kinetics in excised outside-out patches. However, GluK1-1a receptors, when expressed alone, exhibited low peak amplitudes in outside-out recordings that prevented reliable calculation of gating kinetics. Hence, we only compared the properties of receptors co-expressed with Neto proteins, and the results were consistent with those obtained from whole-cell recordings (*Figure 3G-H*, *Table 2*). Neto 2 slowed down desensitization of GluK1-1a by ~1.5 times compared to GluK1-2a ($\tau_{Des}$, GluK1-1a +Neto2: 31.89±4.08 ms; GluK1-2a +Neto2: 20.91±2.11; p=0.0665) (*Figure 3G*, *Table 2*). The desensitization rates for GluK1-1a and GluK1-2a receptors co-expressed with Neto1 was also significantly altered ($\tau_{Des}$, GluK1-1a +Neto1: 4.83±0.46 ms; GluK1-2a +Neto2: 2.84±0.35; *p=0.0310) (*Figure 3G*, *Table 2*). Faster solution exchange times in excised patch recordings also allowed us to measure deactivation kinetics using a 1ms application of 10 mM glutamate. Surprisingly, unlike desensitization, for receptors coexpressed with Neto2, the deactivation rate of GluK1-1a is significantly faster compared to that of GluK1-2a ($\tau_{Dea}$, GluK1-1a +Neto2: 5.18±0.65 ms; GluK1-2a +Neto2: 10.74±1.48; **p=0.0077). In contrast, Neto1 did not significantly alter deactivation kinetics of both GluK1 variant receptors ($\tau_{Dea}$, GluK1-1a +Neto1: 2.83±0.20 ms; GluK1-2a +Neto1: 2.14±0.37; p=0.2086) (*Figure 3H*, *Table 2*). Thus, both whole-cell and excised patch recordings confirm the unique functional properties and differential modulation by Neto proteins due to the presence of fifteen amino acid inserts in GluK1-1a receptors.

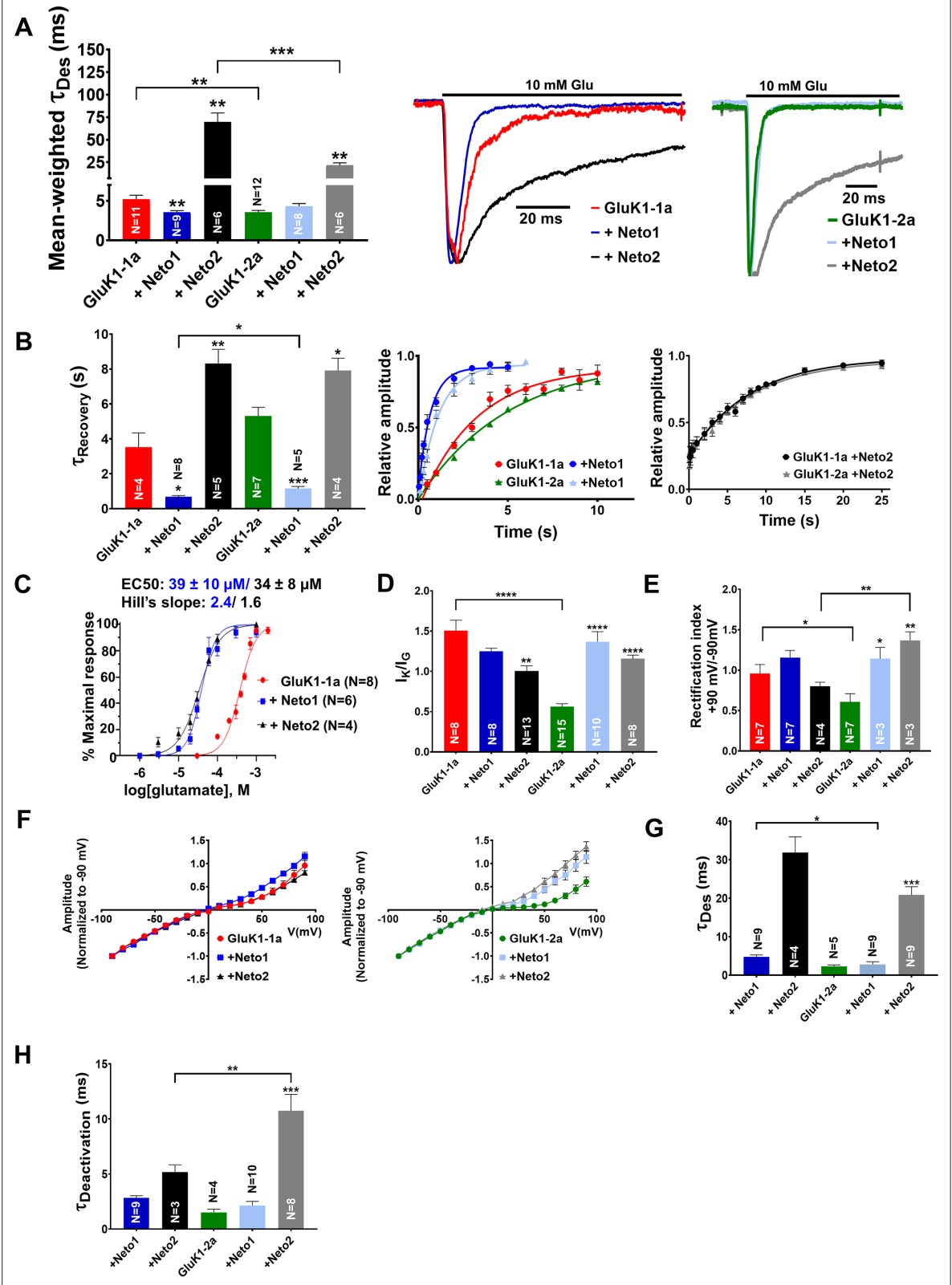

**Figure 3.** Amino-terminal domain (ATD) splice affects the functional modulation of GluK1 kainate receptors by Neto proteins. (**A**) Shows mean-weighted Tau ($\tau_{Des}$) values calculated at 100 ms for GluK1-1a (red) and GluK1-2a (green), respectively, with full-length Neto1 (blue/light blue) or Neto2 (black/gray), in the presence of glutamate. Representative normalized traces are shown for 100 ms application of 10 mM glutamate for HEK293 cells co-expressing GluK1-1a or GluK1-2a with Neto1 and Neto2. (**B**) Shows Tau ($\tau_{Recovery}$) values plotted for GluK1-1a and GluK-2a, respectively, with full-length

*Figure 3 continued on next page*

*Figure 3 continued*

Neto1 or Neto2. Relative amplitude graphs for each receptor in the absence or presence of Neto proteins are also depicted. (**C**) Demonstrates the glutamate dose-response curves for GluK1-1a with Neto proteins. (**D**) Indicates the ratio of peak amplitudes evoked in the presence of 1 mM kainate and 10 mM glutamate for GluK1-1a or GluK1-2a with or without Neto proteins. (**E**) The ratio of currents evoked by the application of 10 mM glutamate at +90 mV and –90 mV for the receptors in the absence or presence of Neto proteins is shown. (**F**) Shows representative IV plots for GluK1-1a and GluK1-2a for the receptor alone versus with Neto proteins, respectively. Panels (**G**) and (**H**) show data recorded from outside-out pulled patches. (**G**) Displays desensitization kinetics for GluK1-1a (red) and GluK1-2a (green) with or without Neto proteins, respectively. (**H**) Shows deactivation kinetics at 1 ms for GluK1-1a (red) and GluK1-2a (green) with or without Neto proteins. Error bars indicate mean ± SEM, N in each bar represents the number of cells used for analysis, and * indicates the significance at a 95% confidence interval.

The online version of this article includes the following source data and figure supplement(s) for figure 3:

**Source data 1.** Data used for the electrophysiology plots.

**Figure supplement 1.** Effect of Neto proteins on agonist-evoked responses of GluK1 receptors.

**Figure supplement 1—source data 1.** Data used for the electrophysiology plots.

## Mutations in GluK1-1a splice insert alter channel properties

Since our electrophysiological analysis showed functional differences between the two splice variants and their modulation by Neto proteins, we created receptors with mutated splice residues and conducted functional assays to identify the key residues. The splice insert (KASGEVSKHLYKVWK) is dominated by positively charged residues and contains four lysines and one histidine. We hypothesized that these charged residues might affect the interactions at the ATD-LBD interface and influence receptor functions. To investigate this, we prepared charge-reversal (K/H to E) and charge-neutral (K/H to A) mutants and carried out functional assays (*Figure 4—source data 2*). Our cell surface biotinylation assay showed that all mutants (glutamate or alanine) reached the cell surface efficiently (data not shown). However, the charge-neutral mutants (K/H to A) gave either very low peak amplitudes (<40 pA) or were not functional and hence, were not included in the study. The charge reversal mutants $K_{368}$-E and $K_{375/379/382}H_{376}$-E revealed fascinating insights into the role of splice residues in altering GluK1 receptor properties (*Figure 4A*). Interestingly, both mutants, $K_{368}$-E and $K_{375/379/382}H_{376}$-E exhibited a significantly slower rate of glutamate-evoked desensitization compared to wild-type GluK1-1a ($\tau_{Des}$, GluK1-1a: 5.21±0.50 ms; $K_{368}$-E: 7.89±1.14 ms *p=0.0334; $K_{375/379/382}H_{376}$-E: 9.62±1.47

**Table 2.** Excised patch outside-out electrophysiology of GluK1-1a and GluK1-2a in the absence or presence of Neto1 (green) or Neto2 (peach).

Errors are reported as SEM. Statistical-significance is reported at 95% CI. p<0.05 (*), p<0.01 (**), p<0.001 (***), p<0.0001 (****) for comparisons between GluK1-1a and GluK1-2a receptors in presence of Neto proteins. Red p-values are statistical significance at 95% for comparison between GluK1-1a and GluK1-2a with either of the Neto proteins.

| Name of the construct | Deactivation | | Desensitization | | Total No. of cells tested |
|---|---|---|---|---|---|
| | $_{Deact}$ (ms) | Rise Time (ms) | $_{Des}$ (ms) | Rise Time (ms) | |
| GluK1-1a | Low peak amplitude | 0.68±0.19 (N=3) | Low peak amplitude | 0.48±0.1 (N=3) | 10 |
| +Neto1 | 2.83±0.2 (N=9) | 0.75±0.04 (N=9) (p=0.7615) | 4.83±0.46 (N=9) | 0.71±0.04 (N=9) | 9 |
| +Neto2 | 5.18±0.65 (N=3) | 1.5±0.24 (N=3) (p=0.0627) | 31.89±4.08 (N=4) | 1.87±0.06 (N=4) | 4 |
| GluK1-2a | 1.51±0.28 (N=4) | 0.58±0.06 (N=4) | 2.34±0.35 (N=5) | 0.72±0.07 (N=5) | 9 |
| +Neto1 | 2.14±0.37 (N=10) (p=0.2086) | 0.75±0.06 (N=10) (p=0.0731) | 2.84±0.69 (N=9) (p=0.5312 / *p=0.0310) | 0.69±0.07 (N=9) | 12 |
| +Neto2 | 10.74±1.48 (N=8) (***p=0.0004 / **p=0.0077) | 1.47±0.23 (N=8) (p=0062) | 20.91±2.11 (N=9) (***p=0.0003 / p=0.0665) | 1.42±0.25 (N=9) | 9 |

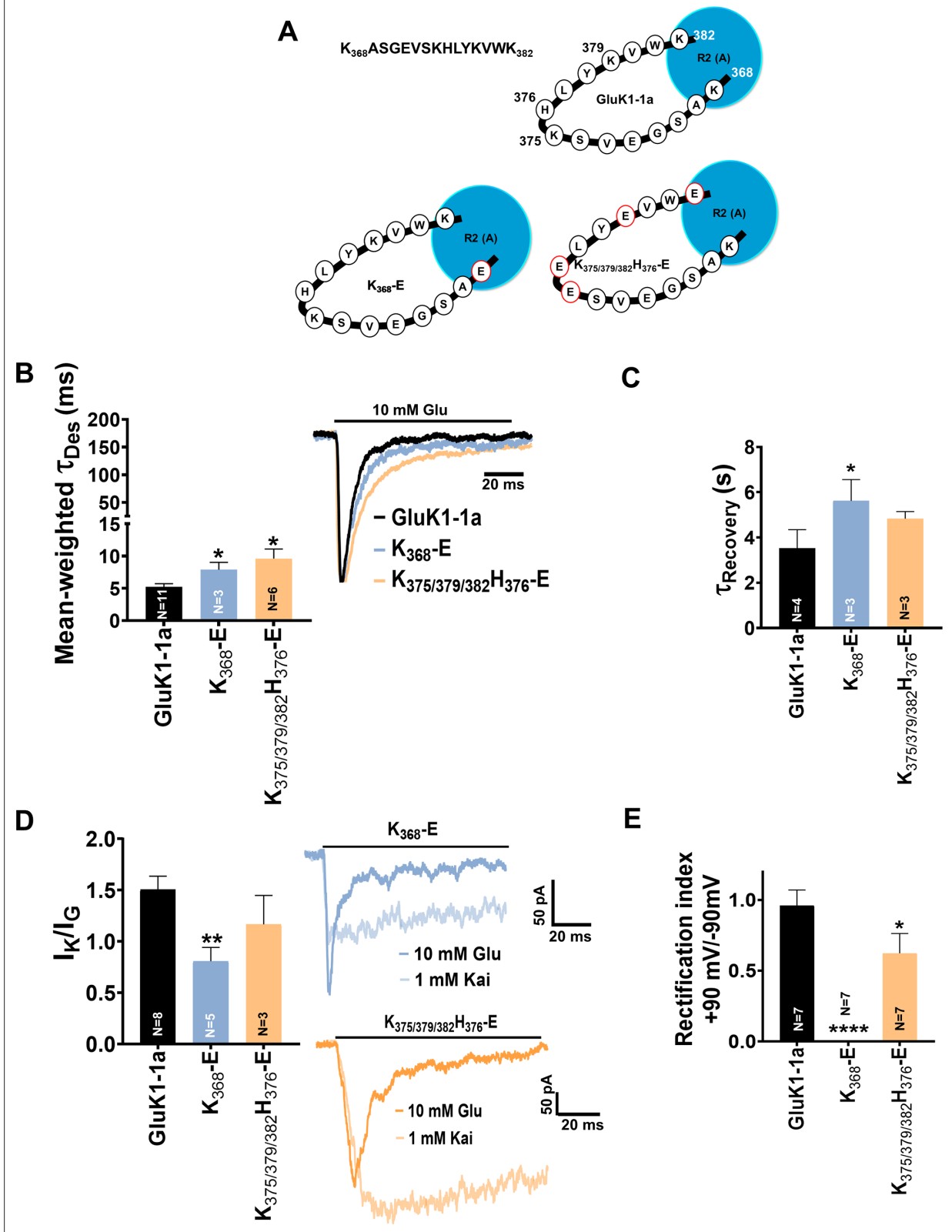

**Figure 4.** Mutation of GluK1-1a splice insert residues affects the desensitization and recovery kinetics of the receptor. Bar graphs (mean ± SEM) show a comparison between wild-type and mutant receptors for different kinetic properties. (**A**) Schematic representation of 15 residues amino-terminal domain (ATD) splice (K$_{368}$ASGEVSKHLYKVWK$_{382}$) in wild-type and mutant receptors under study (**B**) Mean-weighted Tau ($\tau_{Des}$) values for GluK1-1a wild-type and mutant receptors in the presence of 10 mM glutamate. (**C**) Tau ($\tau_{Recovery}$) recovery values for GluK1-1a and mutants. (**D**) The ratio of the peak amplitudes

*Figure 4 continued on next page*

*Figure 4 continued*

evoked in the presence of 1 mM kainate and 10 mM glutamate is shown for GluK1-1a mutants.(**E**) The rectification index represented by the ratio of currents evoked by 10 mM glutamate application at +90 mV and –90 mV for the wild-type and mutant receptors is shown. The wild-type GluK1 splice variant data is the same as from ***Figure 2A*** and is replotted here for comparison. Error bars indicate mean ± SEM, N in each bar represents the number of cells used for analysis, and * indicates the significance at a 95% confidence interval.

The online version of this article includes the following source data for figure 4:

**Source data 1.** Data used for the electrophysiology plots.

**Source data 2.** GluK1-1a amino-terminal domain (ATD) splice mutants.

*p=0.0290) (***Figure 4B***; ***Table 1***). We also observed a significant delay in the recovery from the desensitized state for $K_{368}$-E mutant ($K_{368}$-E: 5.61±0.95 s *p=0.0417) compared to wild-type GluK1-1a. In addition, the $K_{375/379/382}H_{376}$-E mutant also exhibited a slowdown in the recovery, though not significant. ($K_{375/379/382}H_{376}$-E: 4.83±0.31 s p=0.2774) (***Figure 4C***; ***Table 1***). Our investigations of glutamate- and kainate-evoked responses for wild-type and mutant receptors considering their peak amplitudes ($I_K/I_G$) revealed a significant decrease for the $K_{368}$-E mutant (GluK1-1a: 1.51±0.13; $K_{368}$-E: 0.81±0.13 **p=0.0037) and a reduction was observed for $K_{375/379/382}H_{376}$-E receptors (1.17±0.28 p=0.3733) compared to wild-type although the differences do not reach statistical significance (***Figure 4D***; ***Table 1***). These observations are reciprocal to the effect of splice insert compared to the non-spliced form, indicating the importance of these residues in influencing receptor desensitization and recovery. A similar trend reversal was also observed for the measurements of the rectification index for these mutants at positive and negative potentials (+90 mV and –90 mV). The rectification index was significantly reduced in the case of mutant $K_{375/379/382}H_{376}$-E ($K_{375/379/382}H_{376}$-E: 0.62±0.14 *p=0.0499). Surprisingly, no outward rectification was observed for the $K_{368}$-E mutant, and further investigation is needed to fully understand the reasons for the same (***Figure 4E***; ***Figure 5—figure supplement 2C***; ***Table 1***).

## GluK1-1a splice residues $K_{368}$, $K_{375}$, $H_{376}$, $K_{379}$, and $K_{382}$ influence receptor modulation by Neto proteins

Next, we investigated the effects of splice mutants on receptor modulation by Neto proteins. To test whether mutations in splice residues could disrupt interactions with Neto proteins, we performed receptor pull-downs using an antibody against the His-tag of the receptor. Our results showed that the mutant receptors could efficiently pull down Neto1 (detected using the Neto1 antibody) and Neto2-EGFP (detected using the GFP antibody) (***Figure 5—figure supplement 1***), suggesting that mutants don't altogether abolish GluK1-1a and Neto interactions.

Furthermore, we conducted electrophysiology experiments to investigate whether coexpression of Neto proteins can restore functionality and influence the functions of splice mutants. We observed that mutant $K_{368}$-E desensitizes significantly slower in the presence of Neto1 while the mutants $K_{375/379/382}H_{376}$-E and $K_{368/375/379/382}H_{376}$-E do not exhibit any significant deviation ($T_{Des}$, GluK1-1a +Neto1: 3.56±0.22 ms; $K_{368}$-E +Neto1: 4.37±0.25 ms *p=0.0393; $K_{375/379/382}H_{376}$-E +Neto1: 3.74±0.20 ms p=0.5390; $K_{368/375/379/382}H_{376}$-E +Neto1: 4.86±0.67 ms p=0.1242) (***Figure 5A***; ***Figure 5—figure supplement 2A***; ***Table 1***). This observation suggests that some splice residues influence GluK1-1a modulation by Neto1, and more mutational, functional, and structural studies on this interaction are necessary. On the other hand, Neto2 does not significantly affect the desensitization of these mutant receptors compared to wild-type-Neto2 (***Figure 5A***; ***Figure 5—figure supplement 2A***; ***Table 1***).

Similarly, $K_{368}$-E and $K_{368/375/379/382}H_{376}$-E mutants recover significantly faster from the desensitized state when coexpressed with Neto1. On the other hand, the recovery from the desensitized state for mutant receptors is not significantly affected by Neto2. Thus, while Neto1 seems to affect the mutant receptor recovery from the desensitized state, Neto 2 doesn't show significant differences compared to wild-type receptors, again highlighting the differential modulation of GluK1-1a receptors by Neto proteins (***Figure 5B***; ***Table 1***).

Furthermore, to determine whether the agonist efficacy of mutant receptors changed in the presence of Neto proteins, $I_K/I_G$ ratios were measured. Neto1 increased the agonist efficacy for the $K_{368}$-E and $K_{375/379/382}H_{376}$-E mutants but not for $K_{368/375/379/382}H_{376}$-E receptors ($K_{368}$-E +Neto1: 6.99±0.47 ***p=0.0002; $K_{375/379/382}H_{376}$-E +Neto1: 1.69±0.15 *p=0.0326; $K_{368/375/379/382}H_{376}$-E +Neto1: 1.15±0.20 p=0.6495) (***Figure 5C***; ***Figure 5—figure supplement 2B***; ***Table 1***). Similarly, Neto2 also considerably

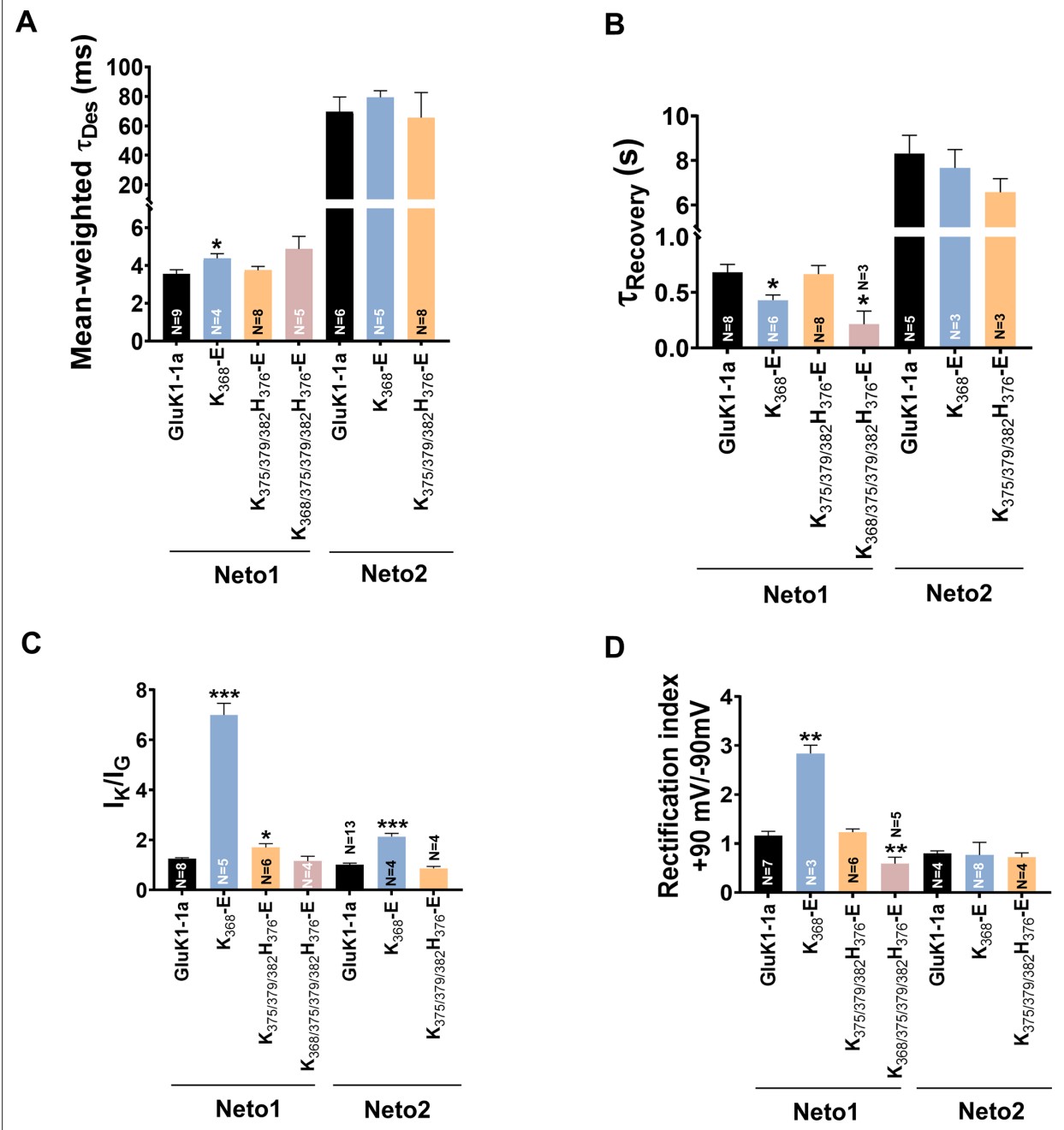

**Figure 5.** Mutation of GluK1-1a splice insert residues affects the receptor modulation by Neto proteins. Bar graphs (mean ± SEM) show a comparison between wild-type and mutant receptors with Neto proteins for different kinetic properties. (**A**) Mean-weighted Tau ($\tau_{Des}$) values for GluK1-1a wild-type and mutant receptors in the presence of 10 mM glutamate and expressed with Neto1/2. (**B**) Tau ($\tau_{Recovery}$) recovery values for GluK1-1a and mutants with Neto1/2. (**C**) The ratio of the peak amplitudes evoked in the presence of 1 mM kainate and 10 mM glutamate for GluK1-1a mutants co-expressed with Neto1/2 is shown. (**D**) The rectification index represented by the ratio of currents evoked by 10 mM glutamate application at +90 mV and –90 mV for the wild-type and mutant receptors with Neto proteins is shown. The wild-type GluK1 splice variants' data is the same as in *Figure 2* and is replotted here for comparison. Error bars indicate mean ± SEM, N in each bar represents the number of cells used for analysis, and * indicates the significance at a 95% confidence interval.

The online version of this article includes the following source data and figure supplement(s) for figure 5:

**Source data 1.** Data used for the electrophysiology plots.

**Figure supplement 1.** Co-immunoprecipitation analysis of GluK1-1a splice mutants and Neto proteins.

**Figure supplement 1—source data 1.** PDF files containing labelled western blots for *Figure 5—figure supplement 1A*.

*Figure 5 continued on next page*

increased the $I_K/I_G$ values for mutant $K_{368}$-E, rescuing the kainate efficacy ($K_{368}$-E +Neto2: 2.13±0.13 ***p=0.0007), while it does not seem to have any significant effect on the $K_{375/379/382}H_{376}$-E mutant ($K_{375/379/382}H_{376}$-E +Neto2: 0.86±0.08 p=0.2045) (*Figure 5C*; *Figure 5—figure supplement 2B*; *Table 1*).

Consistent with the observation above, our examination of the rectification index revealed a significant increase in outward current for the $K_{368}$-E mutant ($K_{368}$-E +Neto1: 2.84±0.16 **p=0.0021). However, $K_{375/379/382}H_{376}$-E did not show any difference compared to wild-type receptors, while $K_{368/375/379/382}H_{376}$-E displayed a significant decrease in the outward current ($K_{375/379/382}H_{376}$-E +Neto1: 1.23±0.07 p=0.5351; $K_{368/375/379/382}H_{376}$-E+Neto1: 0.59±0.13 **p=0.0081) (*Figure 5D*; *Figure 5—figure supplement 2C*; *Table 1*). Interestingly, however, the splice residue mutants did not show any significant variation in the rectification index when coexpressed with Neto2 (*Figure 5D*; *Figure 5—figure supplement 2C*; *Table 1*).

Thus, our analysis of the gating properties of GluK1-1a mutants co-expressed with Neto proteins suggests that positively charged residues at positions 368, 375, 376, 379, and 382 in the splice insert influence receptor modulation by Neto proteins. Neto1 appears to have more pronounced effects on the mutant receptors compared to Neto2. Specifically, Neto1 significantly slowed desensitization for the $K_{368}$-E mutant, accelerated recovery from desensitization for $K_{368}$-E and $K_{368/375/379/382}H_{376}$-E mutants, increased agonist efficacy for $K_{368}$-E and $K_{375/379/382}H_{376}$-E mutants, and altered rectification properties for $K_{368}$-E and $K_{368/375/379/382}H_{376}$-E mutants. In contrast, Neto2 had fewer significant effects on the mutant receptors, with the main impact being an increase in agonist efficacy for the $K_{368}$-E mutant. Notably, Neto2 did not significantly affect desensitization, recovery from desensitization, or rectification properties of the mutant receptors when compared with wild-type GluK1-1a coexpressed with Neto2. These findings suggest that the splice residues in GluK1-1a differentially influence receptor modulation by Neto1 and Neto2, with Neto1 showing more extensive modulation of the mutant receptors' functional properties.

## The structure of GluK1-1a$_{EM}$ shows an overall conserved architecture of the desensitized state in kainate receptors

To evaluate the effects of splice residues on domain organization and structure of GluK1-1a receptors, we pursued its structure determination *via* single particle cryo-EM. Construct optimization was carried out to improve the expression and stability of purified protein. Briefly, the free cysteines in the TM1 region were mutated ($C_{552}Y$, $C_{557}V$) based on the sequence analysis with kainate and AMPA receptors, and this construct was named GluK1-1a$_{EM}$(*Figure 6*, *Figure 6—figure supplements 1 and 2*). The whole-cell patch clamp showed that GluK1-1a$_{EM}$ was functional (*Figure 6—figure supplement 3*).

The structures of GluK1-1a$_{EM}$ were determined using single-particle cryo-EM. The receptors were either DDM solubilized or reconstituted in lipid nanodiscs and trapped in a desensitized state using 2 mM of high-affinity agonist 2 S, 4R-4-methyl glutamate (SYM2081) (*Figure 6—figure supplement 4*). A resolution of ~5.2 Å was achieved for the receptors in lipid nanodiscs, but the transmembrane region was not resolved due to an orientation bias (*Figure 6*; *Figure 6—figure supplement 5*; *Table 3*). The full-length receptor in detergent micelles had a resolution of 8.2 Å, including the transmembrane region, which was ~8 Å for the extracellular domain (*Figure 6—figure supplement 5*; *Table 3*).

A tetrameric receptor model was built based on the crystal structures of GluK1-1a ATD, GluK1 LBD (kainate-bound state; PDB:3C32), and the TM domain based on a highly identical GluK2$_{EM}$ (PDB:5KUF). Our cryo-EM map represented ATD residues-1–398, but the density corresponding to the ATD splice (368-382) was poorly resolved. The ATD-LBD linkers were resolved for all subunits (A to D) in both structures, the S1 and S2 domains were built entirely, and the TMD (TM1, TM3, and TM4) was built only for detergent-solubilized receptors (*Figure 6*; *Figure 6—figure supplements 6 and 7*). For receptors reconstituted in lipid nanodiscs, we observed only the TM3 bundle. TM2 was not

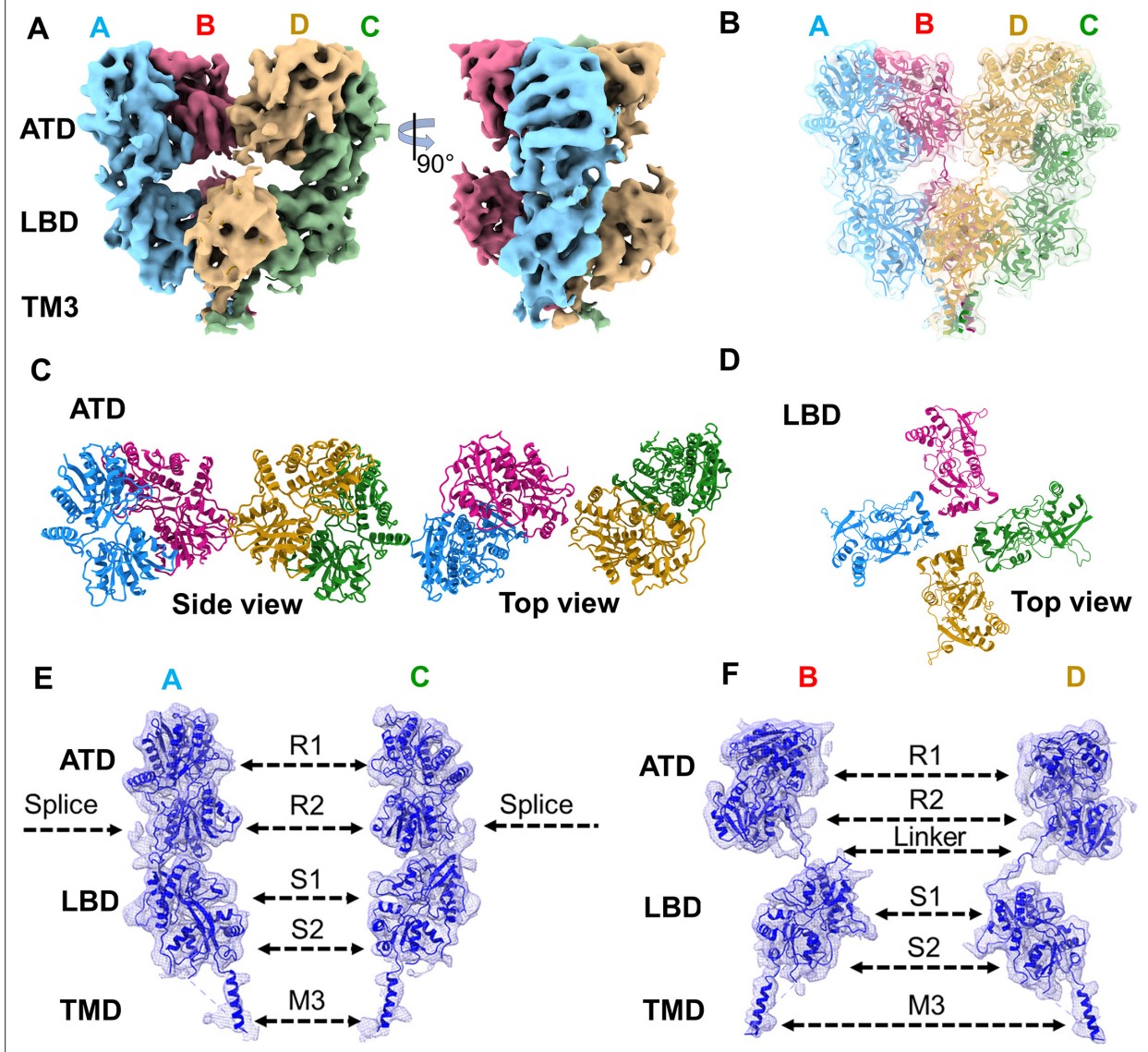

**Figure 6.** Architecture of GluK1-1a$_{EM}$ reconstituted in nanodisc for SYM-bound desensitized state. (**A**) Shows the segmented density map colored according to unique chains of the receptor tetramer (A- blue, B-pink, C-green, and D-gold) at 5.23 Å in side view and 90° rotated orientations. (**B**) Shows the final model fitted in the EM map. (**C** and **D**) Top views of amino-terminal domain (ATD) and ligand binding domain (LBD) layers. (**E** & **F**) Display the segmented map fitted with the corresponding distal (A & C) and proximal (B & D) chains. Receptor sub-domains, the position of splice insertion, and linkers are indicated.

The online version of this article includes the following source data and figure supplement(s) for figure 6:

**Figure supplement 1.** Sequence alignment and construct design of GluK1-1a$_{EM}$.

**Figure supplement 2.** GluK1-1a$_{EM}$ construct design and purification.

**Figure supplement 2—source data 1.** Size exclusion chromatography data used for the plots.

**Figure supplement 2—source data 2.** PDF files containing original SDS-PAGE gels for *Figure 6—figure supplement 2B* -inset 1 with rectangle indicating the cropping margin.

**Figure supplement 2—source data 3.** Original uncropped SDS-PAGE gels for *Figure 6—figure supplement 2B* -inset 1.

**Figure supplement 2—source data 4.** PDF files containing original SDS-PAGE gels for *Figure 6—figure supplement 2B* -inset 2 with rectangle indicating the cropping margin.

**Figure supplement 2—source data 5.** Original uncropped SDS-PAGE gels for *Figure 6—figure supplement 2B* -inset 2.

**Figure supplement 3.** GluK1-1a construct optimization for structural studies and its gating properties.

**Figure supplement 4.** Single-particle cryo-EM data processing flow chart for GluK1-1a$_{EM}$ in nanodisc (ND) and detergent (DDM).

*Figure 6 continued on next page*

*Figure 6 continued*

**Figure supplement 3—source data 1.** Uncropped western blot with rectangle indicating the cropping margin.

**Figure supplement 3—source data 2.** Original file for western blot displayed in *Figure 6—figure supplement 3A*.

**Figure supplement 3—source data 3.** Data used for the electrophysiology plots in panels B-D.

**Figure supplement 5.** Estimation of resolution and particle distribution for GluK1-1a_EM structures.

**Figure supplement 5—source data 1.** Data for the FSC plots.

**Figure supplement 6.** Sequence alignment for the three models presented in the study.

**Figure supplement 7.** Cryo-EM map and model for GluK1-1a_EM dodecyl-β-maltoside (DDM) FL-SYM complex.

**Figure supplement 8.** EM density map labeled to show predicted N-linked glycosylation sites (NXT) for GluK1-1a_EM.

**Figure supplement 9.** Comparison between GluK1-1a_EM (detergent-solubilized or reconstituted in nanodiscs) and GluK1-2a (PDB-7LVT) in the desensitized state.

resolved in either dataset (*Figure 6*; *Figure 6—figure supplement 7*). Extra densities were observed in the ECD layer of the GluK1-1a_EM ND map, which coincided with potential N-linked glycosylation sites (*Figure 6—figure supplement 8*). GluK1-1a_EM maps showed general conservation of the architecture of kainate receptors captured in the desensitized state (*Khanra et al., 2021*; *Kumari et al., 2019*; *Meyerson et al., 2016*; *Selvakumar et al., 2021*). Consistent with earlier studies on homomeric and heteromeric KARs in the desensitized state, GluK1-1a exhibited a modular organization with three layers, namely, the twofold symmetric ATD, quasi-fourfold symmetric LBD and fourfold symmetric TMD. The presence of the ATD splice insert in GluK1-1a did not affect the arrangement of the receptor domains in the desensitized state (*Figure 6*; *Figure 6—figure supplement 7*). A superimposition of GluK1-1a_EM (detergent-solubilized or reconstituted in nanodiscs) and GluK1-2a (PDB:7LVT) showed an overall conservation of the structures in the desensitized state. No significant movements were observed at both the ATD and LBD layers of GluK1-1a with respect to GluK1-2a (*Figure 6*; *Figure 6—figure supplement 9*).

## Discussion

Alternative splicing is a well-known mode of protein function modulation in various ion channels, such as transient receptor potential (TRP) (*Gracheva et al., 2011*; *Zhou et al., 2013*), big potassium (BK) (*Chen et al., 2005*), acid-sensing sodium (ASICs) (*Bässler et al., 2001*), and Shaker K$^+$ channels (*Hoshi et al., 1991*). In the case of iGluRs, the 38 amino acid residue preceding TM4, known as flip and flop isoforms, is a well-characterized module of AMPARs that affects receptor expression and gating kinetics and is involved in various pathophysiological conditions (*Park et al., 2016*; *Sommer et al., 1990*; *Stine et al., 2001*). For kainate receptors, the combination of different subunits (GluK1-GluK5) and post-transcriptional modifications, such as RNA editing and splicing, provides a broader range of pharmacological and gating properties, which can affect synaptic physiology (*Kumar et al., 2011*; *Kumar and Mayer, 2010*; *Lerma, 2006*; *Straub et al., 2016*). GluK1 is the most diversified subunit of KARs due to multiple post-transcriptional events and has been widely studied as a potential drug target. A recent report highlighted the equivalent presence of GRIK1-1 (exon 9) with respect to GRIK1-2 (lacking exon 9) (*Herbrechter et al., 2021*). The dominant presence of the GRIK1-1 gene was also reported in retinal Off bipolar cells of ground squirrels (*Lindstrom et al., 2014*). Interestingly, an N-terminal splice (exon 5) in a similar position has been observed in the GluN1 subunit of NMDA receptors, which is known to influence the interactions at the ATD-LBD interface and, therefore, proton sensitivity and channel kinetics (*Regan et al., 2018*). However, this is the first report to decipher the role of the N-terminal splice insert in the kainate receptor family.

Our study compared the role of ATD splice in the kinetics of the GluK1-1a to a previously extensively characterized equivalent receptor without the splice region, GluK1-2a (*Copits et al., 2011*; *Fisher, 2015*; *Fisher and Fisher, 2014*; *Sommer et al., 1992*; *Swanson and Heinemann, 1998*). We found that the splice insert affects desensitization, recovery from desensitization, glutamate sensitivity, and channel rectification of the GluK1 receptor. Moreover, the splice also had a profound impact on desensitization with kainate, suggesting that the receptor follows different functional pathways for the two agonists. Our results showed a significant deviation from GluK1-2a in kainate- versus glutamate-evoked currents ($I_K/I_G$), implying a role for splice residues in imparting

**Table 3.** Cryo-EM data collection, refinement, and validation for GluK1-1a$_{EM}$.

| | GluK1-1a- 2 S,4R-4-methyl glutamate | | |
|---|---|---|---|
| | GluK1-1a$_{EM}$ ND | GluK1-1a$_{EM}$ DDM ECD | GluK1-1a$_{EM}$ DDM FL |
| **Data Collection and Processing** | | | |
| Microscope | Titan Krios | Titan Krios | |
| Voltage (keV) | 300 | | 300 |
| Number of micrographs | 1535 | | 1100 |
| Camera | K2 | Falcon3 | |
| Mode of recording | Super resolution with energy filter (20 eV slit) | Counting | |
| Exposure time (s) | 12 | | 60 |
| Total dose (e$^-$/Å$^2$) | 40.8 | | 19.5 |
| Defocus range (μm) | 1.8–3.2 | 2.0–3.2 | |
| Pixel size (Å) | 1.41 | | 1.38 |
| Symmetry | C1 | C1 | |
| Initial particle number | 1,97,908 | | 13,750 |
| Final particle number | 24531 | | 5372 |
| Map resolution (Å) | 5.23 | 8.01 | 8.2 |
| FSC threshold | 0.143 | 0.143 | 0.143 |
| **Refinement (Phenix)** | | | |
| Initial model used (PDB code) | (ATD), 3 C32 (LBD), 5KUF(TM3) | (ATD), 3 C32 (LBD) | (ATD), 3 C32 (LBD), 5KUF (TMD) |
| Model resolution (Å) | 5.1/7.3 | 7.2/9.0 | 7.8/9.1 |
| FSC threshold | 0.143/0.5 | 0.143/0.5 | 0.143/0.5 |
| Map-to model fit, CC_mask | 0.71 | 0.69 | 0.73 |
| **Model composition** | | | |
| Non-hydrogen atoms | 21556 | 20880 | 23616 |
| Protein residues | 2684 | 2596 | 2948 |
| **R.m.s. deviations** | | | |
| Bond lengths (Å) | 0.004 | 0.003 | 0.004 |
| Bond angles (°) | 0.861 | 0.796 | 0.805 |
| **Validation** | | | |
| MolProbity score | 1.92 | 2.03 | 2.12 |
| Clashscore | 15.03 | 19.06 | 20.96 |
| **Ramachandran plot** | | | |
| Favored (%) | 96.39 | 96.27 | 95.59 |
| Allowed (%) | 3.53 | 3.65 | 4.38 |
| Disallowed (%) | 0.08 | 0.08 | 0.03 |
| Rotamer outliers (%) | 0.3 | 0.17 | 0.12 |
| Cß outliers (%) | 0.04 | 0 | 0.04 |

higher efficacy for kainate. This likely explains the enhanced stability of the kainate-bound state in GluK1-1a, leading to slower desensitization compared to GluK1-2a. Our functional assays of GluK1-1a receptors co-expressed with Neto proteins showed that Neto1 facilitates faster recovery from the desensitized state, consistent with previous reports (*Copits et al., 2011*; *Fisher, 2015*; *Palacios-Filardo et al., 2016*). However, we did not observe the fast onset of desensitization previously reported for GluK-2a (*Copits et al., 2011*; *Fisher, 2015*). The differences in observations could be due to variations in experimental conditions, such as the constructs and recording conditions used.

Additionally, our results show that Neto2 retards recovery from desensitization for both GluK1 variants, contradicting earlier reports of conflicting recovery rates with Neto2 for GluK1-2a. This indicates that Neto1 and Neto2 may interact with both GluK1 variants in a mutually exclusive manner or that the splice position may regulate the modulation behavior of Neto2 and Neto1. Our comparison of kainate and glutamate efficacies for the two variants also showed differential modulation by Neto proteins. Both Neto1 and Neto2 reduced the $I_K/I_G$ ratios in GluK1-1a, while they significantly increased the $I_K/I_G$ ratios in GluK1-2a, highlighting the impact of the splice insert on receptor response to the agonists (*Figure 3D*; *Figure 3—figure supplement 1B*; *Table 1*).

Our mutational analysis showed that $K_{368}$, a splice insert residue, affects channel kinetics. The charge reversal mutant ($K_{368}$-E) showed significant differences from the wild-type GluK1-1a for all tested properties, indicating a role for $K_{368}$ in protein-protein and/or protein-glycan interactions at the ATD-LBD interface.

Our electrophysiological assays reveal a complex interplay between GluK1 receptor splice variants and Neto auxiliary proteins. While Neto2 demonstrates a more profound impact on wild-type GluK1-1a function compared to GluK1-2a, particularly in slowing desensitization, Neto1's modulation appears more sensitive to specific mutations in the splice insert. This suggests that Neto2 may have a stronger overall effect on receptor function through potentially allosteric mechanisms, whereas Neto1 might interact more directly with specific splice insert residues. The differential sensitivity to mutations highlights distinct roles for these auxiliary proteins: Neto2 as a powerful modulator of overall receptor function, and Neto1 as having more specific interactions with the splice insert region. This complexity underscores the nuanced nature of these protein interactions and the need for further structural and functional studies to fully elucidate the mechanisms underlying the functional diversity of GluK1 receptors.

Although our cryo-EM map poorly resolved the splice region, its position can be ascertained close to the ATD-LBD interface. Based on our functional assays, the splice may possibly affect the interaction between the receptor and auxiliary proteins. The modulatory effects of Neto1 and Neto2 on GluK1 splice variants might be mediated by multiple conserved positively charged patches (*Li et al., 2019*; *Vinnakota et al., 2021*). The complex between GluK2-Neto2 (*He et al., 2021*) provides a model that suggests $K_{183}$ and $K_{187}$ of GluK1 can potentially interact with a negative patch on Neto1 ($D_{140}$-$E_{144}$) and Neto2 ($D_{144}$-$E_{148}$). However, the splice insert appears to be positioned away from this interacting surface. Thus, it is possible that the splice residues may interact with Neto proteins when the receptor adopts a different conformation during the gating cycle. Additionally, splice residues could have an indirect or allosteric influence on the modulation of receptor functions and regulation by Neto proteins.

Our research, which encompasses structural, biochemical, biophysical, and functional investigations, highlights the significance of the unique N-terminal splice insert in the functions of GluK1 kainate receptors and opens avenues for further studies to understand its physiological effects. We also elucidated the important residues within the splice insert that could impact the modulatory behavior of auxiliary proteins. Our study emphasizes the need to investigate all possible combinations of KAR splice variants to better appreciate their contributions at different developmental stages. This comprehensive understanding of the distribution and functional diversity is essential for a rational therapeutic approach involving kainate receptors.

## Materials and methods

### Key resources table

| Reagent type (species) or resource | Designation | Source or reference | Identifiers | Additional information |
|---|---|---|---|---|
| Gene (*Rattus norvegicus*) | GRIK1-1a, GRIK1-2a, Neto1, Neto2 | This paper | | GRIK1 was used with mutations in the TM1 region to improve protein expression and stability |
| Strain, strain background (*Escherichia coli*) | BL21(DE3) | Sigma-Aldrich | CMC0016 | competent cells |
| Cell line (*Homo sapiens*) | HEK293 GnTI-suspension-adapted cells | ATCC | CRL-3022 | Used for expression of GluK1-1aEM for large-scale purification |
| Cell line (*Homo sapiens*) | HEK293 WT cells | ATCC | CRL-1573 | Used for whole-cell patch-clamp electrophysiology |
| Cell line (*Homo sapiens*) | HEK293-T/17 cells | ATCC | CRL-11268 | Used for outside-out patch-clamp electrophysiology |
| Antibody (Rabbit monoclonal) | Anti-His monoclonal antibody | Cell Signaling Technology | Cat. No. 12698 | Used for co-immunoprecipitation/Western Blotting |
| Antibody (Rabbit polyclonal) | Anti-Neto1 polyclonal antibody | Sigma-Aldrich | SAB3500679 | Used for co-immunoprecipitation/Western Blotting |
| Antibody (Mouse Monoclonal) | Anti-GFP | Sigma-Aldrich | G1546 | Used for co-immunoprecipitation/Western Blotting |
| Antibody (Mouse Monoclonal) | Anti-Actin | Sigma-Aldrich | A3853 | Used for co-immunoprecipitation/Western Blotting |
| Chemical compound | SYM2081 (2 S, 4R-4-methyl glutamate) | Tocris Bioscience | 31137-74-3 | Used to stabilize the receptor and electrophysiology experiments |
| Chemical compound | UBP301 | Tocris Bioscience | 569371-10-4 | Used to stabilize the receptor and electrophysiology experiments |
| Chemical compound | Kainic acid | Tocris Bioscience | 487-79-6 | Used to stabilize the receptor and electrophysiology experiments |
| Chemical compound | L-Glutamic acid | Sigma-Aldrich | 49449 | Used to stabilize the receptor and electrophysiology experiments |
| Chemical compound | Sodium Butyrate | Sigma-Aldrich | 8.17500 | Added to boost protein production |
| Commercial assay kit | Bio-Beads SM-2 | Bio-Rad | 1523920 | Used for detergent removal during nanodisc reconstitution |
| Recombinant DNA reagent | pEGBacMam vector | Eric Gouaux's lab (shared) | | Used for protein expression |
| Recombinant DNA reagent | GluK1-1a and GluK1-2a in pRK7 vector | Mark Mayer's lab (shared) | | Used for electrophysiology experiments |
| Software algorithm | cryoSPARCv3 | Nature Methods *Punjani et al., 2017* | DOI:10.1038/nmeth.4169 | Used for single-particle data processing |
| Software algorithm | UCSF Motioncor2 | Nature Methods *Zheng et al., 2017* | DOI 10.1038/nmeth.4193 | Used for motion correction of cryo-EM data |
| Software, algorithm | UCSF ChimeraX | *Goddard et al., 2018* | RRID:SCR_015872 | Molecular |
| Software, algorithm | UCSF Chimera | *Pettersen et al., 2004* | RRID:SCR_004097 | Molecular |
| Software, algorithm | Coot | *Emsley and Cowtan, 2004* | RRID:SCR_014222 | Protein Model |
| Software, algorithm | Phenix | *Adams et al., 2010* | RRID:SCR_014224 | Protein Model |
| Software, algorithm | Clampfit | Molecular Devices | 11.2 | Electrophysiology data analysis |
| Software, algorithm | Fitmaster | HEKA Elektronik | v2x90.4 | Electrophysiology data analysis |
| Software, algorithm | Dotmatics | GraphPad Prism | version 8.0.1 | Used for statistical analysis and graphs/plots |
| Sequence-based reagent | Primers | This paper | PCR primers | Sequences given in *Figure 4—source data 2* |
| Other | Soybean polar lipids | Avanti Polar Lipids | 541602 P | Used for the reconstitution of GluK1-1aEM in nanodiscs |

| Reagent type (species) or resource | Designation | Source or reference | Identifiers | Additional information |
|---|---|---|---|---|
| Other | Dodecyl-β-maltoside (DDM) | Anatrace | D310LA | Used for solubilization of membrane fractions |
| Other | TALON Cobalt Resin | Clontech Takara | 635653 | Used for IMAC purification |
| Other | Protein A Agarose Beads | Thermo Scientific | 20334 | Used for pull-down assays |
| Other | cOmplete Protease Inhibitor Cocktail | Roche (Merck) | 11697498001 | Added to buffers used for protein purification |

HEK293 WT, HEK293 GnTI⁻S, and HEK293-T/17 cells were procured from the American Type Culture Collection (ATCC, USA), accompanied by an authentication certificate and a mycoplasma-free certificate to ensure their quality and integrity.

## Whole-cell patch-clamp electrophysiology

A whole-cell patch-clamp analysis was performed to understand the functional differences between GluK1-1a and GluK1-2a, as well as their modulation by Neto proteins. HEK293 WT mammalian cells were seeded on siliconized glass coverslips in 35 mm dishes using Dulbecco's Modification of Eagle's Medium (DMEM) containing 10% fetal bovine serum (FBS), 2 mM glutamine, and 10 units/mL Penicillin-Streptomycin 24 hr before transection. Cells were transfected with rat (*Rattus norvegicus*) *GRIK1-1a* or *GRIK1-2a* cloned in pRK7 (in the presence or absence of r*Neto1*/r*Neto2* cloned in IE-pRK8 as indicated) with EGFP expressing plasmid using Xfect (Clontech) according to manufacturer's instructions (receptor: Neto in 1: 3 and Receptor: EGFP in 1:1 DNA concentration). Similar protocols were followed to test the functionality of the *GRIK1-1a*$_{EM}$ and *GRIK1-1a* splice mutant constructs. The whole-cell patch-clamp recording was performed with an EPC 10 USB amplifier (HEKA) 24–48 hr post-transfection. Cells were lifted using 1.5 mm diameter thin wall glass capillary tubes (30–0066 Harvard Apparatus), pulled to a fine tip with a Sutter P-1000 micropipette puller (Sutter Instruments, Novato, CA) containing internal solution (10 mM HEPES pH 7.4, 100 mM CsF, 30 mM CsCl, 4 mM NaCl, 0.5 mM CaCl$_2$, 5 mM EGTA and 300 mOsm). Cells were continuously perfused with external/bath solution (10 mM HEPES pH 7.4, 150 mM NaCl, 2.8 mM KCl, 1.8 mM CaCl$_2$, 1 mM MgCl$_2$, and 300 mOsm) through a peristaltic perfusion system (Multichannel systems). Current was measured by holding the membrane at –60 mV, using a 2 kHz low-pass filter. Ligands (10 mM glutamate, 1 mM kainate, 2 mM SYM2081, or 10 µM UBP301) were applied through a double-barrelled theta pipette connected with an ultra-fast piezo-based perfusion system (Multichannel systems) *via* v8 Perfusion Fast-Step System (VC-77SP). Recordings were controlled and measured using Patchmaster-v2x90.2 (Heka Elektronik). The raw files were analyzed using Clampfit 11.2 (Molecular Devices) and Fitmaster-v2x90.4 (HEKA Elektronik). The data was fitted to logistic curves in GraphPad Prism (GraphPad Software Inc). All traces were normalized before being used for calculations. The single exponential two-term fitting (Levenberg-Marquardt) was used to estimate the rate of desensitization ($\tau_{Des}$) for 100ms glutamate application, measured as the decline of the current from 80% of its peak amplitude. Mean-weighted Tau ($\tau_{Des}$) values were determined using the formula [($\tau$1×amplitude1) + ($\tau$2×amplitude2)]/[amplitude1 +amplitude2] where the tau values are the time constants from the exponential fit and the amplitudes are the assessed contributions of each component to the total peak current amplitude. To compute the recovery from desensitization, a series of paired-pulse experiments were performed with varying time pulses, where the amplitudes of the test pulse were normalized to that of the desensitizing pulse (calculated as relative amplitude) and plotted in comparison to the time (in seconds) between these two pulses. The recovery rate ($\tau_{Recovery}$) was obtained by fitting the one-phase association with an exponential function. Dose-response experiments were performed for GluK1-1a (co-expressed with or without Neto proteins) and GluK1-2a with different concentrations of glutamate or kainate in the range of 1 µM to 3 mM or 0.5 µM to 600 µM, respectively. Dose-response values were calculated as the percentage of maximum response against log[agonist] concentrations and fitted using variable slope (Hill's equation) in GraphPad Prism 8.0.1. For calculating kainate efficacy ($I_K/I_G$), the ratio of peak amplitudes obtained from the same cell evoked first by glutamate followed by kainate were employed. A ratio of peak amplitudes obtained at +90/–90 mV was utilized to calculate the rectification index. We investigated the voltage-dependent endogenous polyamine block by measuring current-voltage relationships for the wild-type GluK1 receptors in the absence or presence of Neto

proteins. Current-voltage (IV) plots were prepared using the voltage ramp from –90 to +90 mV with a 10 mV increment step only after complete recovery of the receptor, and the current amplitude was normalized to that obtained at –90 mV. For the mutant receptors, current values were obtained only for 3 voltage steps, –90, 0, and +90 mV. For kainate-evoked currents, the percentage desensitization was calculated. The steady-state current measured at the end of 1 s kainate application was divided by peak current. This ratio was then subtracted from 1 and multiplied by 100 to give percentage (%) desensitization (*Fisher and Fisher, 2014*).

## Outside-out patch electrophysiology

HEK 293 T/17 cells were used for outside-out patch experiments 48–96 hr post transfection. Cells were transfected with *GRIK1-1a* or *GRIK1-2a* cloned in pRK7 (in the presence or absence of rat (*Rattus norvegicus*) r*Neto1*/r*Neto2* cloned in IE-pRK8 as indicated) with EGFP expressing plasmid at DNA ratios of 3:0.5 for receptor alone and 2:8:0.5 with Neto proteins, respectively, using TransIT-LT1 Transfection Reagent according to manufacturer's instructions.

The extracellular solution used for the experiment contained 10 mM HEPES pH 7.3, 150 mM NaCl, 2.8 mM KCl, 2 mM $CaCl_2$, 1 mM $MgCl_2$, and 10 mM glucose. The intracellular solution contained 10 mM HEPES pH 7.3, 110 mM CsF, 30 mM CsCl, 5 mM EGTA, 4 mM NaCl, and 0.5 mM $CaCl_2$.

Patch micropipettes were pulled using a P-97 Sutter puller, and the resistance was maintained at 2.3–2.6 megaohms. After forming the whole-cell configuration, the patch was pulled away from the cell to facilitate an outside-out patch. The cells were voltage-clamped at –70 mV, and a saturated concentration of glutamate (10 mM) was rapidly applied using a three-barrelled theta glass attached to a Siskiyou MXPZT-300 solution switcher. Glutamate applications of 1ms were used to determine deactivation, while 100 or 1000 ms applications were used to assess desensitization.

## Statistical analysis

Comparisons between wild-type receptors, EM, or mutant constructs were obtained using an unpaired *t*-test (two-tailed, with or without the Welch test) or Brown-Forsythe and Welch ANOVA, followed by Dunnett's multiple comparisons. Statistical analysis was carried out in GraphPad Prism, version 8.0.1. p-values <0.05 were considered statistically significant and are reported (*p<0.05, **p<0.01, ***p<0.001, ****p<0.0001).

## Site-directed mutagenesis (SDM)

Based on our electrophysiology analysis of the wild-type GluK1-1a/GluK1-2a receptors, structural analysis of GluK1-1a$_{EM}$, and recent reports that suggest that the presence of the positive patches in GluK2 ATD affects the interaction with Neto proteins (*Li et al., 2019*; *He et al., 2021*; *Vinnakota et al., 2021*), we performed SDM to understand the role of splice residues in the receptor kinetics. All splice mutations were introduced in the wild-type (species)*GRIK1-1a* pRK7 construct for electrophysiology, as well as the *GRIK1-1a$_{EM}$*-EGFP-His$_8$-pEGBacMam construct for surface expression and pull-downs using the (overlap PCR) ligation-free cloning approach (*Zhang et al., 2017*). In brief, we performed the two sets of PCR using standard cloning primers (~300 bp upstream and downstream of the mutation) and the mutant primers as listed in *Figure 4—source data 2* to obtain the fragment containing our mutation of interest and flanking regions corresponding to the wild-type GluK1-1a. Next, the obtained fragment was used as a megaprimer to amplify the complete GluK1-1a vector backbone that would now contain the mutation. This PCR product was DpnI digested for 1 hr at 37 °C to remove parental DNA and was transformed into DH5α strain of *E. coli*. Clones were confirmed by sequencing. Initially, the positively charged and other residues of the splice were substituted with alanine. Later, charge reversal mutants (K and/or H to E) were also prepared to understand their role in the receptor kinetics as well as interaction with Neto proteins. The mutants prepared are summarized in *Figure 4—source data 2*.

## Co-immunoprecipitation (in vitro)

To understand the effect of mutations on GluK1-1a and Neto1/2 interaction, co-immunoprecipitation was carried out using an anti-His monoclonal antibody against the receptor. Similar constructs and transfection protocols were followed for surface expression analysis. Cells were pelleted 65–70 h post-transfection and washed with TBS (20 mM Tris pH 8, 150 mM NaCl). These cells were sonicated and

solubilized in 500 uL lysis buffer (20 mM Tris pH 8, 150 mM NaCl, 1% glycerol, protease inhibitor cock-tail, 30 mM DDM). Post solubilization, debris was removed by centrifuging at 17,000 × g for 45 min at 4 °C. The supernatant was incubated for pre-clearing with 20 μL of pre-equilibrated Protein A agarose beads (Thermo Scientific) for 1 hr on a rotator at 4 °C. Post-pre-clearing, ~10% sample was saved as input, and the rest was used for pull-downs. Simultaneously, 2 ug of anti-His antibody (host: rabbit, Cat. No.-12698, Cell Signaling Technology) was added to 40 μL pre-equilibrated Protein A agarose (Thermo Scientific) and incubated for 1 hr at 4 °C, followed by the addition of pre-cleared lysate. It was further incubated at 4 °C overnight (14–16 hr). The unbound fraction was removed and washed four times with 500 μL wash buffer (20 mM Tris pH 8, 150 mM NaCl, 1% glycerol, 0.75 mM DDM) to remove non-specific interactions. Protein was eluted in 30 μL elution buffer (100 mM Tris pH 6.8, 12% glycerol, 4% SDS, 10 mM DTT, 2% β- mercaptoethanol) by heating at 95 °C for 10 min. Rabbit IgG controls were set up to confirm the validity of the experiment. To analyze the pull-down, 8% SDS-PAGE was used, followed by a western transfer. To detect the internal control (actin), the receptor, and the co-immu-noprecipitated Neto proteins, the immunoblots were probed using anti-actin (mouse, A3853, Sigma), anti-His (rabbit, 12698, Cell Signaling Technology), and anti-Neto1 (rabbit, SAB3500679, Sigma) or anti-GFP (mouse, G1546, Sigma) antibodies.

## Construct design for expression and purification of rat GluK1-1a$_{EM}$

To obtain functional GluK1 receptors, rat (*Rattus norvegicus*) *GRIK1* with ATD splice insert and the shortest C-terminal domain, *GRIK1-1a* (1–871 amino acid residues), was cloned in the pEGBacMam vector. The receptor was cloned in-frame with a thrombin recognition site (LVPRGSAAAA), EGFP (A$_{207}$K; non-dimerizing mutant), and His$_8$ at the C-terminus. The wild-type protein generated very low amounts of the tetramer, as observed in fluorescence-assisted size-exclusion chromatography (FSEC). Therefore, based on the alignments of the sequence with GluK2$_{EM}$, GluK3$_{EM}$ and GluA2, we mutated free cysteines in the TM1 region to residues corresponding to those of GluK2$_{EM}$ or GluA2 [1 x Cys (C$_{576}$S), 2 x Cys (C$_{552}$Y, C$_{557}$V) and 3 x Cys (C$_{552}$Y, C$_{557}$V, C$_{576}$S)] to obtain good yields of the tetrameric receptor. All clones were confirmed by restriction digestion and sequencing. The expression of all the mutants was confirmed by immunoblotting against His-tag and FSEC. GluK1-1a with 2 x Cys muta-tions (C$_{552}$Y, C$_{557}$V) gave us the best receptor quality as observed in FSEC and, therefore, was used as GluK1-1a$_{EM}$ for large-scale purification for structural studies.

## GluK1-1a$_{EM}$ expression and purification

Three liters of Human embryonic kidney (HEK) 293 GnTI$^-$ suspension-adapted cultures (~1.5–2.0 × 10$^6$ cells/mL) were transfected (or infected with virus) with rat *GRIK1-1a$_{EM}$* plasmid at 0.5 μg/mL or baculo-virus, prepared in DH10Bac as per established protocol (*Goehring et al., 2014*) using polyethylenei-mine (PEI-MAX, Polysciences; 1 DNA: 3 PEI w/w) as the transfection agent (or, infected at a multiplicity of infection of ~2). To boost the protein production, 10 mM sodium butyrate (Sigma) was added 16 hr post-transfection/infection, and cultures were incubated at 30 °C for protein expression. Cells were harvested 65–70 h after transfection/infection, washed with buffer containing 20 mM Tris pH 8 and 150 mM NaCl, and stored at –80 °C until further processing. The cell pellet was resuspended (20 mL/L culture volume) in lysis buffer (20 mM Tris pH 8, 150 mM NaCl, and protease inhibitor cocktail, Roche). These resuspended cells were disrupted using sonication (QSonica sonicator, three cycles of 90 s; 10 s ON/20 s OFF; temperature cut-off: 15 °C). The lysate was clarified using low-speed spin (4307 × g for 20 min at 4 °C). The membranes were pelleted using ultracentrifugation (118,991 × g for 1 hr at 4 °C). Solubilization of membrane fraction was done in buffer containing 20 mM Tris pH 8, 150 mM NaCl, 30 mM DDM (Anatrace, D310LA), 5% glycerol, 10 mM imidazole, and protease inhibitor cocktail for 1 hr on rotator at 4 °C. The non-solubilized fraction was separated by centrifugation at 47,850 × g for 1 hr at 4 °C and 4 mL of cobalt-charged TALON resin (Clontech, Takara) was added to the solubilized fraction for batch binding (3 hr at 4 °C). Beads were harvested. The column was packed and washed with wash buffer (20 mM Tris pH 8, 150 mM NaCl, 1 mM DDM, 1% glycerol, 40 mM imidazole) until OD$_{280}$ reached zero. The bound receptor was eluted with elution buffer (20 mM Tris pH 8, 150 mM NaCl, 1 mM DDM, 1% glycerol, 250 mM imidazole). The peak fractions were pooled, concentrated at 1.3 mg/mL, and digested overnight at 4 °C with thrombin (1:100 w/w). Simultaneously, purified GluK1-1a$_{EM}$ was also reconstituted in MSP1E3D1 nanodiscs with soybean polar lipids (Avanti Polar Lipids, 541602 P) in a 1:2:140 ratio (GluK1-1a:MSP:lipids) following already established protocols

(*Chen et al., 2017*; *Gao et al., 2016*; *Ritchie et al., 2009*). In brief, the purified GluK1-1a$_{EM}$ in DDM was incubated with lipids, MSP1E3D1, 14 mM sodium cholate, and 1 mM PMSF for 30 min at 4 °C. The reconstitution of protein in nanodisc was initiated by removing detergent using equilibrated Bio-Beads SM-2 (biorad, 150 mg/mL) for 4–6 hr at 4 °C on an end-to-end rotator. The thrombin-digested GluK1-1a$_{EM}$ receptors in DDM, or nanodisc-reconstituted receptors, were further purified via gel filtration (Superose 6 10/300, GE) in buffer containing 20 mM Tris pH 8, 150 mM NaCl, 0.5% glycerol, and 0.75 mM DDM (no detergent for nanodisc protein). The peak fractions were pooled and concentrated to ~0.6 mg/mL (final DDM concentration,~7.5 mM) or ~0.9 mg/mL GluK1-1a$_{EM}$ in nanodisc. All the affinity elution fractions and final purified protein were confirmed for purity and homogeneity using SDS-PAGE and FSEC, respectively. In the subsequent sections, purified GluK1-1a$_{EM}$ in detergent and nanodisc will be called GluK1-1a$_{EM}$ DDM and GluK1-1a$_{EM}$ ND, respectively.

## Cryo-EM sample preparation and data collection

Before grid preparation, the purified protein was incubated with 2 mM 2 S, 4R-4-methyl glutamate (SYM2081) to capture the receptor in the desensitized state. The concentration of SYM2081 was confirmed by electrophysiology, and the stability of the receptor-SYM2081 complex was tested using FSEC.

GluK1-1a$_{EM}$ DDM: Double application of 3 µL protein (~0.6 mg/mL) SYM2081 complex was carried out on glow-discharged gold grids (R 0.6/1, 300 mesh, Quantifoil). The grids were blotted for 4.5 s and 3.5 s, respectively (0 blot force). Vitrobot temperature was maintained at 12 °C with 100% humidity, and the sample was vitrified in liquid ethane. The clipped grids were loaded into a 300 keV Titan Krios microscope equipped with a Falcon III direct detector camera (4k × 4k). Movies were recorded in counting mode with a nominal magnification of 59,000 X (pixel size: 1.38 Å), and the defocus range of −2.0 to −3.2 µm increased in steps of 0.3. Each movie comprises 25 frames with a total exposure of 60 s. A dose rate of 0.78 e⁻/frame was applied, with the total dose being 19.5 e⁻/Å$^2$.

GluK1-1a$_{EM}$ ND: Double application of 2 µL protein (~0.9 mg/mL) SYM2081 complex was carried out on glow-discharged gold grids (1.2/1.3, 200 mesh, Quantifoil). Grids were blotted for 3 s and 10 s, respectively, followed by vitrification. The clipped grids were loaded into a 300 keV Titan Krios microscope equipped with a K2 direct-detector camera (Gatan). The movies were recorded in super-resolution mode with an energy filter (20 eV slit) and a pixel size of 1.41 Å. Each movie was composed of 30 frames with a total exposure of 12 s. A dose rate of 1.36 e⁻/frame was applied, with a total dose of 40.8 e⁻/Å$^2$.

## Single-particle analysis

All movies (GluK1-1a$_{EM}$ DDM-SYM: 1100, GluK1-1a$_{EM}$ ND-SYM: 1535) were motion-corrected using UCSF Motioncor2 (*Zheng et al., 2017*). Bad micrographs were removed manually post-contrast transfer function (CTF) estimation using CTFFIND4 (*Rohou and Grigorieff, 2015*). Manually picked particles (~1000) were 2D classified and used as templates for auto picking in RELION or cryoSPARCv 3. For GluK1-1a$_{EM}$ DDM-SYM and GluK1-1a$_{EM}$ ND-SYM, the auto picked particles in cryoSPARCv3 were cleaned up with multiple rounds of 2D classification. Particles in the best classes were used for training TOPAZ for automated particle picking. For GluK1-1a$_{EM}$ DDM-SYM, initially, 13,750 particles were picked; post iterative rounds of 2D and 3D classification, 5372 particles were used for final 3D reconstruction. In the case of GluK1-1a$_{EM}$ ND-SYM, initially, 1,97,908 particles were picked and subjected to multiple rounds of clean-up using 2D and 3D classification. Finally, 24,531 particles were used for the final 3D reconstruction.

The final particles were corrected for local motion in cryoSPARCv3, followed by refinement using C1 symmetry for both GluK1-1a$_{EM}$ DDM-SYM and GluK1-1a$_{EM}$ ND-SYM. Since we had less information on TMD in both the detergent and nanodisc forms, as observed in 2D classes, we performed local refinement for ATD and LBD using an ECD mask for GluK1-1a$_{EM}$ DDM-SYM and GluK1-1a$_{EM}$ ND-SYM, which improved map resolution to 8.01 Å and 5.2 Å (0.143 FSC), respectively. For GluK1-1a$_{EM}$ DDM-SYM, local refinement using a mask for full-length structure yielded a resolution of 8.2 Å. The 3D maps were sharpened via cryoSPARCv3 (*Punjani et al., 2017*) or DeepEMhancer (*Sanchez-Garcia et al., 2021*) using a mask from the cryoSPARCv3 refinement output. Local Resolution was estimated using BlocRes (*Cardone et al., 2013*) in cryoSPARCv3 or Phenix1.19.2 (*Afonine et al., 2018a*).

## Model building

Tetrameric assembly for the GluK1-1a$_{EM}$ ND-SYM model was built using crystal structures of individual domains (ATD, not yet published, and LBD, PDB: 3C32, GluK1 LBD crystal structure with kainate) fitted into the EM map (5.2 Å) in UCSF Chimera (*Pettersen et al., 2004*). Furthermore, for building GluK1-1a$_{EM}$ ND transmembrane domain 3 and GluK1-1a$_{EM}$ DDM trans-membrane domain, GluK2 TMD (PDB:5KUF) was used due to high identity (~95%) for fitting into EM density. Phenix1.19.2 (*Afonine et al., 2018b*) and Namdinator (*Kidmose et al., 2019*) were used to improve the model via the rigid body and Molecular Dynamics based flexible fitting, respectively. The final models fit well into the EM map of GluK1-1a$_{EM}$ ND-SYM, GluK1-1a$_{EM}$ DDM-SYM ECD, and FL, respectively, with a definite density for ATD, LBD, and TMD (pre-M1, M1, M3, and M4). Coot 0.9.4 (*Emsley and Cowtan, 2004*) was used to analyze the final models, and chimera/chimeraX (*Pettersen et al., 2021*; *Pettersen et al., 2004*) was used for figure preparation.

## Supplementary methods

### Spatiotemporal distribution of exon 9 of GluK1 using transcriptomics data analysis

To understand the abundance of *GRIK1-1* splice in the human brain, we resorted to RNA-seq data from the BrainSpan atlas that constitutes various databases to study transcriptional mechanisms in human brain development. The RNA-seq data were downloaded for *GRIK1* (https://www.brainspan.org/rnaseq/gene/1099967), plotted in Excel (x-axis: regions of the brain; y-axes: log$_2$ transformed normalized expression intensity values in Reads Per Kilobase of exon model per Million mapped reads, RPKM and age of donor). The heat maps were generated using the Excel file (CSV format) and RStudio (https://www.rstudio.com/) to determine the presence of GluK1 in various regions of the brain at different developmental stages, as shown in *Figure 1*. Furthermore, we narrowed it down to the *GRIK1-1* splice (exon 9; start position 30968845, 45 nucleotides in length) and tried to understand how it overlaps with the entire *GRIK1* gene expression. These data explored the presence of the ATD splice irrespective of which C-terminal splice variant is present.

## Acknowledgements

We thank Dr. Mark L Mayer and Dr. Sagar Chittori, National Institute of Child Health and Human Development, National Institutes of Health, Bethesda, USA, for kindly sharing the coordinates and maps of GluK1-1a ATD which were used to generate the tetrameric receptor model. We also thank Dr. Mayer for his critical reading of the manuscript and his feedback. Dr. Eric Gouaux kindly provided the pEG BacMam vector. Access to EM was provided by the National Electron cryo-microscopy facility at the Bangalore Life Sciences Cluster. We thankfully acknowledge the kind assistance of Dr. Vinoth-kumar Kutti Ragunath, NCBS, Bangalore, in grid preparation and EM data collection.

## Additional information

### Funding

| Funder | Grant reference number | Author |
| --- | --- | --- |
| Department of Science and Technology, Ministry of Science and Technology, India | CRG/2020/003971 | Janesh Kumar |

The funders had no role in study design, data collection and interpretation, or the decision to submit the work for publication.

### Author contributions

Surbhi Dhingra, Data curation, Formal analysis, Validation, Investigation, Visualization, Methodology, Writing – original draft, Writing – review and editing; Prachi M Chopade, Formal analysis, Investigation, Writing – original draft; Rajesh Vinnakota, Data curation, Formal analysis,

Investigation, Methodology; Janesh Kumar, Conceptualization, Resources, Formal analysis, Supervision, Funding acquisition, Writing – original draft, Project administration, Writing – review and editing

**Author ORCIDs**
Surbhi Dhingra http://orcid.org/0000-0003-4349-3215
Prachi M Chopade http://orcid.org/0000-0001-5394-1136
Rajesh Vinnakota http://orcid.org/0000-0002-2354-5105
Janesh Kumar https://orcid.org/0000-0003-0767-3788

Reviewer #1 (Public review): https://doi.org/10.7554/eLife.89755.4.sa1
Author response https://doi.org/10.7554/eLife.89755.4.sa2

## Additional files

### Supplementary files
• MDAR checklist

### Data availability

The cryo-EM density reconstructions and final models were deposited in the Electron Microscopy Data Base (accession codes EMD-34076, EMD-34083, and EMD-34197) and the Protein Data Bank (accession codes 7YSJ, 7YSV, and 8GPR).All data generated or analysed during this study are included in the manuscript and supporting files; source data files have been provided.

The following datasets were generated:

| Author(s) | Year | Dataset title | Dataset URL | Database and Identifier |
|---|---|---|---|---|
| Dhingra S, Kumar J | 2023 | GluK1-1a in nanodisc captured in SYM2081 bound desensitized state | https://www.ebi.ac.uk/emdb/EMD-34076 | EMDataBank, EMD-34076 |
| Dhingra S, Kumar J | 2023 | GluK1-1a extracellular domain captured in SYM2081 bound desensitized state | https://www.ebi.ac.uk/emdb/EMD-34083 | EMDataBank, EMD-34083 |
| Dhingra S, Kumar J | 2023 | GluK1-1a receptor captured in the desensitized state | https://www.ebi.ac.uk/emdb/EMD-34197 | EMDataBank, EMD-34197 |
| Dhingra S, Kumar J | 2023 | GluK1-1a in nanodisc captured in SYM2081 bound desensitized state | https://www.rcsb.org/structure/7YSJ | RCSB Protein Data Bank, 7YSJ |
| Dhingra S, Kumar J | 2023 | GluK1-1a extracellular domain captured in SYM2081 bound desensitized state | https://www.rcsb.org/structure/7YSV | RCSB Protein Data Bank, 7YSV |
| Dhingra S, Kumar J | 2023 | GluK1-1a receptor captured in the desensitized state | https://www.rcsb.org/structure/8GPR | RCSB Protein Data Bank, 8GPR |

The following previously published datasets were used:

| Author(s) | Year | Dataset title | Dataset URL | Database and Identifier |
|---|---|---|---|---|
| Lein ES | 2007 | GRIK1 | https://www.brainspan.org/rnaseq/gene/1099967 | Brainspan, 1099967 |

*Continued on next page*

*Continued*

| Author(s) | Year | Dataset title | Dataset URL | Database and Identifier |
|---|---|---|---|---|
| Mayer ML | 2008 | Crystal structure of GluR5 ligand-binding core in complex with sodium at 1.72 Angstrom resolution | https://www.rcsb.org/structure/3C32 | RCSB Protein Data Bank, 3C32 |
| Meyerson JR, Chittori S, Merk A, Rao P, Han TH, Serpe M, Mayer ML, Subramaniam S | 2016 | GluK2EM with 2S,4R-4-methylglutamate | https://www.rcsb.org/structure/5KUF | RCSB Protein Data Bank, 5KUF |
| Meyerson JR, Selvakumar P | 2021 | Structure of full-length GluK1 with L-Glu | https://www.rcsb.org/structure/7LVT | RCSB Protein Data Bank, 7LVT |

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
