## [Editor Report · eLife Assessment]

This **important** study shows that a splice variant of the kainate receptor Glu1-1a that inserts 15 amino acids in the extracellular N-terminal region substantially changes the channel's desensitization properties, the sensitivity to glutamate and kainate, and the effects of modulatory Neto proteins. In the revised paper the authors have clarified several points raised by reviewers but the structural portion of the study has not been improved and consequently, more data are needed to determine the molecular mechanism by which the insert changes the functional profile of the channel. Even so, these **solid** findings advance our understanding of splice variants among glutamate receptors and will be of interest to neuro- and cell-biologists and biophysicists in the field.

---

## [Referee Report · Reviewer #1 (Public review)]

Kainate receptors play various important roles in synaptic transmission. The receptors can be divided into low affinity kainate receptors (GluK1-3) and high affinity kainate receptos (GluK4-5). The receptors can assemble as homomers (GluK1-3) or low-high affinity heteromers (GluK4-5). The functional diversity is further increased by RNA splicing. Previous studies have investigated C-terminal splice variants of GluK1, but GluK1 N-terminal (exon 9) insertions have not been previously characterized. In this study Dhingra et al investigate the functional implications of a GluK1 splice variant that inserts a 15 amino acid segment into the extracellular N-terminal region of the protein using whole-cell and excised outside-out electrophysiology.

The authors convincingly show that the insertion profoundly impacts the function of GluK1-1a - the channels that have the insertion are slower to desensitize. The data also shows that the insertion changes the modulatory effects of Neto proteins, resulting in altered rates of desensitization and recovery from desensitization. To determine the mechanism by which the insertion exerts these functional effects, the authors perform pull-down assays of Neto proteins, and extensive mutagenesis on the insert.

The electrophysiological part of the study is very rigorous and meticulous.

The biggest weakness of the manuscript is the structural work. Due to issues with preferred orientation (a common problem in cryo-EM), the 3D reconstructions are at a low resolution (in the 5-8 Å range) and cannot offer much mechanistic insight into the effects of the insertion. The authors have opted to keep this data unchanged in the revised manuscript.

Despite this, the study is a valuable contribution to the field because it characterizes a GluK1 variant that has not been studied before and highlights the functional diversity that exists within the kainate receptor family.

---

## [Author Response]

The following is the authors’ response to the previous reviews.

We are grateful to all three reviewers and editors for their critical comments and suggestions.

**Reviewer #2 (Recommendations For The Authors):**
The authors responded satisfactorily to all my comments and suggestions.

We thank the reviewer for his time and feedback.

**Reviewer #3 (Recommendations For The Authors):**
Comments for authors:The authors have addressed most of the reviewer's concerns. Although no additional data were included to strengthen the manuscript, they have clarified some relevant points, and the manuscript has been updated accordingly. In my view, the current manuscript is well-written and mostly straightforward.

We thank the reviewer for his time and suggestions. Addressing them have improved the quality of our manuscript.

After a second revision, I just have a few minor comments (mostly editorial) that should be easy to address.(1) Page 16: "The dominant presence of the GRIK1-1 gene was also reported in retinal Off bipolar cells..." Please include reference(s).

We have now cited the following reference:

Lindstrom, S.H., Ryan, D.G., Shi, J., DeVries, S.H., 2014. Kainate receptor subunit diversity underlying response diversity in retinal Off bipolar cells. J. Physiol. 592, 1457–1477. https://doi.org/10.1113/jphysiol.2013.265033

(2) Page 18: "Based on our functional assays, the splice seems to affect the interaction between the receptor and auxiliary proteins". Please remove or tone down this statement; the current data do not support this claim.

We have revised the sentence as following: “Based on our functional assays, the splice may possibly affect the interaction between the receptor and auxiliary proteins.”

(3) Page 24: "cultures ... at 0.5 µg/mL were transfected". In the current context, it is not clear what you mean with 0.5 µg/mL. Please check and correct.

Thanks for pointing out this error. We have corrected it.

(4) Page 30. He et al. reference is repeated.

Thanks. We have fixed it now.

(5) Figure 3, Panel C: Please incorporate the EC50 value for the red trace into the figure; it appears to be a different data set and, consequently, a different fitting compared with Figure 2C.

The GluK1-1a data set (red trace) is identical to that in Figure 2c, though it may appear different due to the scale of the X and Y axis. As suggested, we have now included the EC50 value for this data set in Figure 3, panel C.

(6) Figure legend 4: Please check two minor issues here:(a) "Bar graphs... with or without Neto1 protein..." This statement is apparently wrong; Figure 4 does not show the effect of Neto1.(b) "The wild type GluK1 splice variant data is the same as from Figure 1.." I think the authors mean Figure 2A instead of Fig. 1. Please check.

Thanks for pointing out the error. We have fixed the same in the revised manuscript.

(7) Please check and correct spelling/wording issues in the text. Here are some examples:(a) Page 9 " Figure 3G - I, Table2.." (There is no Panel I).

Fixed.

(b) Page 16 "... and is involved in various pathophysiology..."

We have revised the sentence as “… and is involved in various pathophysiological conditions”

(c) Page 19 "The constructs used for this study were HEK293 WT mammalian cells were seeded on..."

Fixed. Thanks.

(d) Page 23 "The immunoblots were probed..." Please check the whole paragraph and correct the issues.

Fixed. Thanks.

(e) Page 27 "initially, 1,97,908 particles were picked". Check the value; the same issue occurs in Fig.6 table supplement 1.

Thanks. We have now modified the sentence to clarify that for GluK1-1aEM ND-SYM, initially, 1,97,908 particles were picked and subjected to multiple rounds of clean-up using 2D and 3D classification. Finally, 24,531 particles were used for the final 3D reconstruction and refinement.

(f) Legend Figure 2: Remove "(F)" from the legend.

Thanks. Fixed.

(g) Legend Figure 2-Sup.1: Check/correct spelling issues.

Thanks. Fixed.

(h) Figure 5-figure supplement 1: There is a mistake in panel B: "GFP" label is shown for Gluk1 and Neto2, but the authors mention that the pull-down was done with Anti-His antibodies. Please correct.

Thanks. The pull-down experiments were done with anti-His for both the blots presented in panels A and B as mentioned in both the figures (right side panels of both A and B). However, for the GluK1 and Neto2 pull downs (panel B), the blots were probed with anti-GFP antibody which would detect both the receptor (as the receptor has both GFP-His8) and Neto2-GFP at their respective sizes. This has been indicated in the figure panel B.

(8) Related to the point-by-point document:Major concern 2: Interpreting the effect of mutants on the regulation by Neto proteins requires knowing how the mutant is affecting the channel properties without Neto. In my view, if the data showing the K368/375/379/382H376-E mutant without Neto is missing (in this case due to low current amplitude), then, the pink bars in Fig. 5 should be removed from the figure.

We thank the reviewer for raising this interesting point and agree that it would be valuable to characterize the channel properties of all the mutants individually. However, as mentioned earlier, the functions of some mutant receptors are only rescued, or reliable, measurable currents are detected, when they are co-expressed with Neto proteins. We still believe that comparing wild-type and mutant receptors co-expressed with Neto proteins provides important insights, and therefore, we would like to retain the K368/375/379/382H376-E mutant data in the figure.

Major concern 4: Figure 6-figure Supplement 8 is not mentioned in the manuscript. It would help to include a proper description in the Results section similar to the answer included in the point-by-point document.

Figure6-figure Supplement 8 has already been cited on page 15. We have also cited Figure6-figure Supplement 9 on the same page and have added following sentences in the text:

“A superimposition of GluK1-1aEM (detergent-solubilized or reconstituted in nanodiscs) and GluK1-2a (PDB:7LVT) showed an overall conservation of the structures in the desensitized state. No significant movements were observed at both the ATD and LBD layers of GluK1-1a with respect to GluK1-2a (Figure 6; Figure 6-figure supplement 9).”

Major concern 5: The ramp/recovery protocol was not included properly in the manuscript; please include the time of the ramp pulse and the time used for the recovery period.

Elaborated ramp and recovery protocols are included in the methods section. The time used for the recovery period was variable and was tuned as per the recovery kinetics. All the figures were representative traces are shown include the scale bar showing the time period of agonist application.

Minor concern 1: The proposed change was not included in the manuscript; check page 7.

Thanks for highlighting this error. We have now changed it in the revised manuscript.

Minor concern 10: The manuscript was not corrected as indicated. Please check.

Thanks. We have now modified the sentence as following: “…..a reduction was observed for K375/379/382H376-E receptors (1.17 ± 0.28 P=0.3733) compared to wild-type *although differences do not reach statistical significance*”

Minor concern 14: The figure was not corrected as indicated. Please check.

Thanks for highlighting this error. We have now changed it in the revised manuscript.

Minor concern 19: I suggest including this briefly in the Discussion section.

Thanks for the suggestion. We have included the following sentence in the discussion:

“The differences in observations could be due to variations in experimental conditions, such as the constructs and recording conditions used.”